



# Revisiting the tropical Atlantic western boundary circulation
# from a 25-year time series of satellite altimetry data
Djoirka M. Dimoune[1], Florence Birol[2], Fabrice Hernandez[1,2], Fabien Léger[2], Moacyr Araujo[1,3]
[1]Laboratorio de Oceanografia Física Estuarina e Costeira (LOFEC), Departamento de
Oceanografia da Universidade Federal de Pernambuco (UFPE), Cidade Universitária, Avenida
Arquitetura s/n, 50740-550 Recife, PE, Brazil
[2]LEGOS, Université de Toulouse, CNES, CNRS, IRD, 18 avenue Edouard Belin, 31400,
France
[3]Brazilian Research Network on Global Climate Change (Rede CLIMA), Av. dos Astronautas,
1758, 01227-010 São José dos Campos, SP, Brazil
*Correspondance to:* Djoirka M. Dimoune (pmintodimoune@gmail.com)
**Abstract.** Geostrophic currents derived from altimetry are used to investigate the surface
circulation in the Western Tropical Atlantic over the 1998-2017 period. Using six horizontal
sections defined to capture the current branches of the study area, we investigate their respective
variations at both seasonal and interannual time-scales as well as the spatial distribution of these
variations. Our results show that the central branch of the South Equatorial Current, the North
Brazil Current component located south of the equator, the Guyana Current and the northern
branch of the South Equatorial Current at 42° W have similar annual cycles, with
maxima/minima during boreal winter-spring/October-November. In contrast, the seasonal
cycles of the North Brazil Current branch located between the equator and 7-8° N, the North
Brazil Current retroflected branch and the North Equatorial Countercurrent show
maxima/minima during boreal fall/May. West of 42° W, an eastward current is observed
between 0°-2° N, identified as the equatorial extension of the retroflected branch of the North
Brazil Current. It is part of a large cyclonic circulation observed between 0°-6° N and 35°-45°
W during boreal spring. The North Equatorial Countercurrent shows a two-core structure during
the second half of the year, when we also observe the two regions where the North Brazil
Current retroflects. The latter can be related to the wind stress curl seasonal changes. At
interannual scales, depending on which side of the equator, the North Brazil Current exhibits
two opposite scenarios related to the tropical Atlantic Meridional Mode phases. The interannual
variability of the North Equatorial Countercurrent and of the northern branch of the South
Equatorial Current (in terms of both strength and/or latitudinal shift) at 42° W are also
associated to the Atlantic Meridional Mode, while they are associated to the zonal mode phases
at 32° W.



## 1    Introduction

The energetic Western Tropical Atlantic (WTA) boundary surface circulation is known to play a key role in the transport of heat, salt and water mass from the southern to the northern hemispheres of the Atlantic Ocean. It corresponds to the return branch of the thermohaline Atlantic Meridional Overturning Circulation (AMOC), influenced by the wind (Schmitz and McCartney, 1993; Schott et al., 2004; Rodrigues et al., 2007). A regional scheme of the surface currents in the study area is proposed in Fig. 1. It is derived from a global analysis of the different works mentioned below.

From 5° S to 15° N, the surface boundary circulation is formed by the North Brazil Current (NBC) flowing northward along the South American shelf. It carries tropical waters originating from the South Atlantic subtropical gyre and contributes to interhemispheric water transport (Johns et al., 1990; 1998; Peterson and Stramma, 1991; Stramma and England, 1999; Fratantoni et al., 2000; Silva et al., 2009; Zheng and Giese, 2009, Garzoli and Matano, 2011). The NBC has its origin near 5° S, with two sources: the central branch of the westward South Equatorial Current (cSEC); and the along-shelf equatorward North Brazil Undercurrent (NBUC) which rises to the surface around 5-6° S (Schott et al., 1998, Dossa et al., 2020). The latter advects warm waters from the South Equatorial Current (SEC) through its southern branch (Schott et al., 1995). Further north, around 5° N, the NBC is also fed by the northern branch of the SEC (nSEC) (Goes et al., 2005). Then, between 5°-9° N and 45-50° W, a large part of the NBC retroflects to form a southeastward retroflected branch (called hereinafter rNBC). Between 3° N and 8° N, this branch first feeds the eastward North Equatorial Countercurrent (NECC) throughout the year, except during the boreal spring. At that time, the NECC is fed only by the North Equatorial current (NEC) (Bourlès et al., 1999a; Goes et al., 2005). The NECC flows eastward between 2° N and 12° N, and crosses the tropical Atlantic (Didden and Schott, 1992; Ffield, 2005; Urbano et al., 2008; Araujo et al., 2017). During the second half of the year, this current shows two cores that can separate into a southern and a northern branch (called sNECC and nNECC, respectively: Urbano et al., 2006; 2008).

At depth, around 3-8° N, Cochrane et al. (1979) and Schott et al. (2004) suggested that, part of the rNBC also feeds the eastward North Equatorial UnderCurrent (NEUC) located around 5° N. In addition, between 2° S and 3° N, the rNBC feeds the subsurface eastward Equatorial UnderCurrent (EUC) (Hisard and Hénin, 1987; Bourlès et al., 1999b; Hazeleger et al., 2003;



Hazeleger et de Vries, 2003; Schott et al., 1995; 2004). North of 10° N, the part of the NBC
which has not retroflected flows northwestward along the Guyana coast, forming the Guyana
Current. The latter is also fed seasonally by the NEC (Johns et al., 1998) and transports warm
equatorial waters into the Caribbean Sea ( Stramma and Schott, 1999; Garzoli et al., 2003).
The WTA boundary surface circulation is wind-driven. In the vorticity equation, the terms that
dominate localy are the Ekman pumping and the divergence of the geostrophic currents (Garzoli
and Katz, 1983; Urbano et al., 2006). The region is also known to be influenced by large
mesoscale activities due to the barotropic instabilities of the currents (Aguedjou et al., 2019;
Aroucha et al., 2020). Previous studies of Garzoli and Katz (1983), Jochum and Malanotte-
Rizzoli (2003) and Verdy and Jochum (2005) about the WTA boundary circulation and the
NECC indicate that, west of 32° W, the Sverdrup balance is no more respected; the advection
terms of the relative vorticity due to the eddies and the mean flow become then important. North
of the equator, the region is characterized by strong seasonal variability of the wind. The Trade
Winds variations influence the current system formed by the NBC, the NBC retroflection
(NBCR), the rNBC and the NECC. In particular the NBCR location, the NBC transport and the
NECC position/transport respond to the seasonal changes in the wind regimes (Johns et al.,
1990; 1998; Garzoli et al., 2003; 2004; Urbano et al., 2006; 2008). This wind influence is
traduced by a latitudinal changes of the currents, in conjunction with the Intertropical
Convergence Zone (ITCZ) location. The variability of the current strength appears as a regional
response of the wind stress curl (WSC) distribution and of the WSC strength over the basin
(Johns et al., 1998; Fonseca et al., 2004; Garzoli et al., 2004, Urbano et al., 2006; 2008). In the
equatorial region, the EUC seasonal variability is first associated to the basin scale zonal
pressure gradient (ZPG), and also to the seasonal cycle of the local wind forcing (Hisard and
Hénin, 1987; Provost et al., 2004; Brandt et al., 2006; Hormann and Brandt, 2007; Brandt et
al., 2016).
The interannual variability of the WTA boundary currents has been little studied because of the
lack of long-term data in this area. Nevertheless, Fonseca et al. (2004), using a combination of
altimetry and hydrographic data from 1993 to 2000, investigated the influence of the wind on
both the NBCR and the NECC variability. They did not find any direct relationship between
them. Hormann et al. (2012), used the surface velocity data derived from drifters between 1993
and 2009 and highlighted a relationship between the NECC intensity/location and the tropical
Atlantic climate modes (ACM), represented by positive and negative phases of the Atlantic
zonal mode (AZM) and the Atlantic meridional mode (AMM) (Cabos et al., 2019). In the





equatorial Atlantic, Hormann and Brandt (2007) also found such relationship, using a high-
resolution ocean general circulation model, observations and sea surface temperature (SST)
data. They showed that the EUC transport is affected by the cold and warm events of the AZM
(the so called "Atlantic Niño / Niña") and confirmed the previous findings of Goes and Wainer
(2003) concerning the link between the interannual variability of the wind and the ACM
impacting the strength of the tropical Atlantic circulation.
In this study, we propose to revisit the scheme of the WTA boundary surface circulation using
a 25-year time series of gridded altimeter-derived geostrophic currents. This dataset is longer
than the one used by Fonseca et al. (2004) and allows to provide a more robust description of
the current branches described above, as well as of their seasonal and interannual variations.
The data also allows to infer regional relationship among these currents. The paper is organized
as follow: in Sect. 2, the data and methods used are presented. Section 3 brings some general
characteristics of the current variability in the study area. In the fourth section we analyze and
discuss the seasonal and spatial variabilities of the surface geostrophic currents and propose an
updated seasonal map of the WTA surface circulation. The interannual variability of the
circulation is analyzed in Sect. 5. Section 6 is devoted to a general discussion, and Sect. 7 offers
a summary and some perspectives.








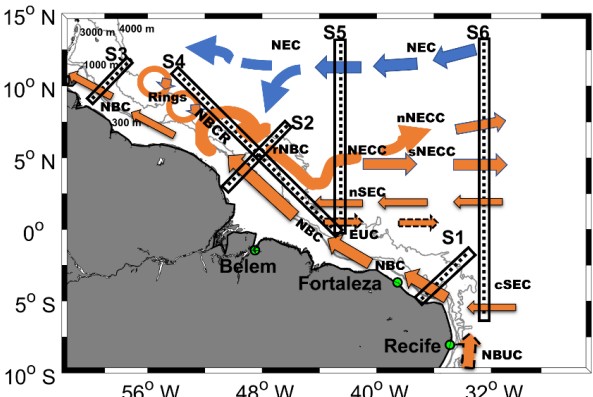

**Figure 1.** Schematic view of the western boundary surface circulation in the tropical Atlantic derived from Schott et al. (2004), Goês et al. (2005), Urbano et al. (2006; 2008) and Aroucha et al. (2019). The distribution of the horizontal section used to study the different current branches are also indicated in black: S1, S2, S3, S4, S5 and S6. Solid and dashed arrows are the upper and the subsurface currents, respectively. The blue and orange colors of the arrows show connections with the northern and southern hemisphere waters, respectively. From south to north, the currents shown are the North Brazil Undercurrent (NBUC), the central and northern branches of the South Equatorial Current (cSEC and nSEC, respectively), the North Brazil Current, its retroflection and its retroflected branch (NBC, NBCR and rNBC, respectively), the Equatorial Undercurrent (EUC), the North Equatorial Countercurrent (NECC) and its southern and northern branches (sNECC and nNECC, respectively), and the North Equatorial Current (NEC). The 300 m, 1000 m, 3000 m and 4000 m isobaths (grey lines) are from the ETOPO2v1 database.





## 2    Data and methods

### 2.1    Altimeter-derived geostrophic currents

From along-track altimetry sea surface height measurements of all available satellite missions, the Copernicus Marine Environment Monitoring Service (CMEMS) produces daily maps of ocean dynamic topography, and derives then geostrophic surface currents. Here, we use the SEALEVEL_GLO_PHY_L4_REP_OBSERVATIONS_008_047 product (https://resources.marine.copernicus.eu/) from January 1993 to December 2017. Daily maps of dynamic topography are estimated by optimal interpolation on $0.25° \times 0.25°$ global grid (details can be found in Pujol et al., 2016), and the geostrophic currents are computed using the 9-points stencil width methodology (Arbic et al., 2012) for latitudes outside the equatorial band (Equator $\pm5°$) and the β-plane approximation (Lagerloef et al., 1999) in the equatorial band.

For the present work, focusing on the seasonal and interannual variability, the daily gridded velocity fields from CMEMS have been averaged on a monthly basis. Note that we have removed data located in the Amazon region because they show unrealistic values, probably due to erroneous altimetry measurements which have not been correctly flagged. We have then defined six horizontal sections (called S1, S2, S3, S4, S5 and S6, respectively), so that they cross perpendicularly at least one of the regional current branches (see Fig. 1). For each section, the original zonal and meridional surface velocity components have been rotated in order to derive the along-section and cross-section velocity components. In this study, we considered only the cross-section component.

### 2.2    Ekman currents

We also used the GEKCO product (Geostrophic and Ekman Current Observatory, Sudre et al., 2013), made available by LEGOS (Laboratoire d'Etudes en Géophysique et Océanographie Spatiale). It provides the zonal and meridional components of the daily wind-driven currents at a 0.25° resolution. The latter were calculated using the standard Ekman formulation, and the estimates have been improved in the equatorial region. Then, they were validated with independent observations from both Lagragian and Eulerienne perspectives (see Sudre et al. 2013 for more details). For this work, the daily current estimates have been monthly averaged over the period 1993-2017.

### 2.3    Wind velocity

Monthly wind velocity fields from the ERA5 atmospheric reanalysis produced by the European Centre for Medium-Range Weather Forecasts (ECMWF, http://www.ecmwf.int) are used in



order to evaluate the influence of the remote winds on the WTA ocean circulation. They were
downloaded from the Copernicus Climate Change data server over the January 1993 -
December 2017 period. We used the wind velocity data to calculate the wind stress field as
follows.
The zonal and meridional components of the wind stress, $\zeta_x$ and $\zeta_y$ are calculated using empirical
formulations (Large and Pound; 1981: Gill, 1982; Trenberth et al., 1990) following NRSC

(2013):

$\zeta_x = \rho_{air} C_D W * U$                    (1)

$\zeta_y = \rho_{air} C_D W * V$                    (2)

where U/V represent the zonal/meridional wind velocity components; W, the wind speed
amplitude; $\rho_{air}$, the air density (1.2 kg m$^{-2}$); $C_D$, the drag coefficient at the ocean surface,
calculated according to Large and Pound (1981).
The wind stress curl (WSC) is then deduced following Gill (1982):
                  $$Curl(\zeta) = \frac{\partial \zeta y}{\partial x} - \frac{d\zeta x}{dy}$$                    (3)

From the WSC estimates, we compute the ITCZ location (~zero values location), the
minimum/maximum of the negative/positive values, and the WSC strength (sum of the absolute
minimum negative value and the absolute maximum positive value) in the tropical Atlantic.
Each of these parameters is zonally averaged over the region covering 6° S -16° N, and 30° W-
0° W, following Fonseca et al. (2004).
**2.4   Sea Surface temperature**
Monthly estimates of Sea Surface temperature (SST) are also used in order to compute the
Atlantic climate mode indexes and evaluate their possible relationship with the interannual
changes observed in the WTA boundary circulation. A global gridded SST product, with a 1°
spatial       resolution       is       downloaded       from       the       NOAA       repository
(https://www.esrl.noaa.gov/psd/gridded/data.noaa.oisst.v2.html, Reynolds et al. 2002).
The AZM index is calculated considering the SST anomalies (SSTA) relative to the 1993-2017
monthly climatology in the ATL3 region bounded by 3° S-3° N/20° W-0° E (Zebiak, 1993,
Hormann et al., 2012). The AMM index is also based on SSTA relative to the 1993-2017
monthly climatology, and calculated as the difference between the spatial average SSTA in the



box 5° N-25° N/60° W-20° W and the spatial average SSTA in the box 20° S-5° N/30° W-10°
E (Servain, 1991; Hormann et al., 2012).

## 3    General characteristics of the circulation in the Western Tropical Atlantic

The mean WTA surface geostrophic circulation is first derived by averaging the gridded
altimetry current maps over 1993-2017 (Fig. 2a). We distinguish three different areas. From
south/east to north/west:
1)  The NBC formation area starting around 5°S: the westward cSEC flowing north of 6°S

(mean value of ~ 0.3 m s$^{-1}$) feeds the NBC-NBUC current system around 34°-36° W.

The NBC amplitude increases along its northward along-shelf course, up to 0.8 m s$^{-1}$

around 3° S. Then it slows down toward the equator, before increasing again north of

3° N. Along the Guyana coast, its mean velocities are again weaker, with values of ~

0.3 m s$^{-1}$.

2)  The NBC retroflection region between 5°-8° N: in this area, the NBC undergoes an

eastward recirculation, which feeds a southeastward vein of current so-called the NBC

retroflected branch (rNBC). The rNBC reaches annual mean velocities of ~0.6 m s$^{-1}$.

3)  The area located between 3°-6° N and 42°-46° W: it represents the region where the

rNBC meanders with an annual mean velocity of 0.5 m s$^{-1}$ to partly feed the surface

eastward NECC which decays along its course. This area is located in the region of high

wind variability (Figure not shown).

In addition, we also observe the westward nSEC flowing between 2°-6° N, with stronger
velocity values at the eastern part of the basin (mean velocity larger than 0.3 m s$^{-1}$).
We computed the mean power spectral density of the daily geostrophic current time series in
order to detect the dominant components of the WTA current variations (not shown). It
highlighted three main energy zones:
1)  at intraseasonal scales with different peaks at periods less than 120 days,
2)  at seasonal scale,
3)  at interannual timescale, with peaks at periods larger than 600 days.
We filtered the velocity time series using different cutoff frequencies in order to isolate each of
this component of the current variability: below 120 days, between 120 days and 600 days and
above 600 days. Then, we computed the ratio between the standard deviation of each filtered





current field and of the total current field. The resulting maps (Fig. 2b-d) show the relative
importance of each component with regard to the total variance, as a function of the location.
We observe the predominance of the seasonal variability in the whole WTA (overall ratio of
0.44), with the highest values (0.48) observed along the continental shelf, in the NBC region,
and between 0°-2° S east of 36° W. Intraseasonal fluctuations are also important in the same
areas (with largest ratio of 0.44) while the interannual variability is only noticeable north/east
of 4° N/40° W, where the NECC is located, with values representing less than 0.2 of the total
variance (consistent with Richardson and Walsh, 1986). In this study, we focus on the seasonal
and interannual timescales.















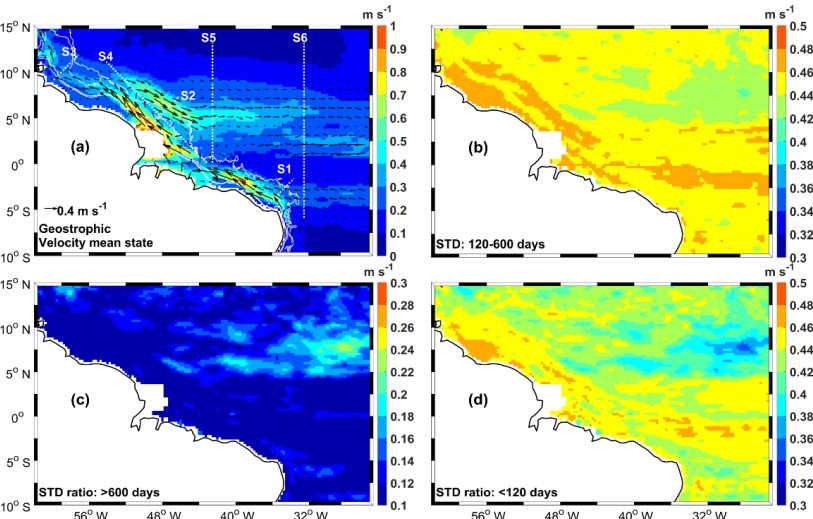

**Figure 2. (a)** Temporal mean of the geostrophic currents (amplitude in m s⁻¹ and vectors) in the study area between 1993 to 2017; **(b), (c)** and **(d)** ratios between the standard deviations of the currents for the signals between 120 and 600 days, more than 600 days and less than 120days, respectively, and the standard deviation of the total currents. The white dashed lines S1, S2, S3, S4, S5, S6 in **(a)** represent the cross-sections of the currents, and the solid white lines are the 300 m, 1000 m, 3000 m and 4000 m isobaths.



Then we monthly averaged the daily geostrophic currents and extracted the cross-section
geostrophic velocities along the six sections defined above (Fig. 1). In order to remove the
intraseasonal variability, the monthly velocity estimates are further low-pass filtered using a 4-
month cutoff frequency. The time-space diagrams (also called Hovmöller diagrams) along the
six sections are plotted in Fig. 3. We can clearly observe large changes in both time and space
with respect to the different current branches, illustrating the complexity of the surface
circulation in the WTA. The seasonal and, in a lesser extent, the interannual current fluctuations
mentioned above are clearly visible in the different sections. For each of them, the time-average
current values are also computed as a function of the latitude (left-side plots in Fig. 3),
highlighting the mean spatial extension of the current veins crossing the corresponding section.
Note that in addition to the main currents mentioned in Fig. 1, between 1.5° S-2° N, the sections
4 to 6 show an eastward current, positioned where the EUC might be located. At 44° W, Bourlès
et al. (1999b) have also noticed the presence of a surface eastward flow above the EUC, which
was identified to be different than the EUC. So, we chose to name it X hereinafter in order to
further investigate this signal and its origin.
Table 1 summarizes the mean current width derived along the different sections (using left-side
plots in Fig. 3). Note first that a positive current convention is chosen for the sections as follows:
northward NBC along sections 1 to 3; eastward NBCR along section 4; eastward NECC along
sections 5 and 6, and eastward X flows along sections 4 to 6. Hence, the signatures of the rNBC
on the section 2, the nSEC (sections 4 to 6) and the cSEC (section 6) are considered as negative.
From Table 1, we observe that the NBC becomes narrower from the section 1 to the section 2.
The retroflection zone (section S4, Fig. 3S4) extends from 3.7° N to 10.5° N, in agreement with
Fonseca et al., (2004) who found the northernmost position of the NBCR around 11° N. North
of the retroflection, the NBC along the Guyana coast is weaker, but broader relatively to the
section 2 (NBC2 and NBC3 in Fig. 3). From section 6 to 4, the nSEC signature changes. It is
wider at 32° W (nSEC6), then narrower at 42° W (nSEC5), and widens again closer to the shelf
at 44° W (nSEC4) (respectively 550, 67 and 190 km in Table 1). The NECC extension also
varies from 32° W to 42° W, and is wider and located further north on the east, with a mean
width extending from 860 km (NECC5) to 920 km (NECC6).







**Table 1.** Extension in latitude and km of the different currents branches crossing the different sections: NBC1,
NBC2 and NBC3 are respectively the North Brazil Current captured on sections 1, 2 and 3; NECC5 and NECC6
are respectively the North Equatorial Countercurrent on the sections 5 and 6; nSEC4, nSEC5 and nSEC6 are
respectively the northern branch of the South Equatorial Current (SEC) on the sections 4, 5 and 6; cSEC6 is the
central branch of the SEC on the section 6; and X4, X5 and X6 are the equatorial eastward flow X on respectively
on the sections 4, 5 and 6.

| List of current veins | Latitudinal coverage | Current width (km) |
|---|---|---|
| NBC1 | 1.3° S-4.6° S | ~ 520 |
| NBC2 | 3.6° N-5° N | ~ 220 |
| rNBC2 | 5° N-7.4° N | ~ 380 |
| NBC3 | 9.1° N-11.9° N | ~ 440 |
| NBCR4 | 3.7° N-10.5° N | ~ 760 |
| nSEC4 | 2° N-3.7° N | ~ 190 |
| X4 | 1.2° N-2° N | ~ 120 |
| NECC5 | 2.4° N-10.1° N | ~ 860 |
| nSEC5 | 1.7° N-2.3° N | ~ 70 |
| X5 | 0° N-1.7° N | ~ 190 |
| NECC6 | 4.1° N-12.4° N | ~ 920 |
| nSEC6 | 0° N-4.1° N | ~ 550 |
| X6 | 0° S-1.4° S | ~ 110 |
| cSEC6 | 1.4° S-5.9° S | ~ 500 |
























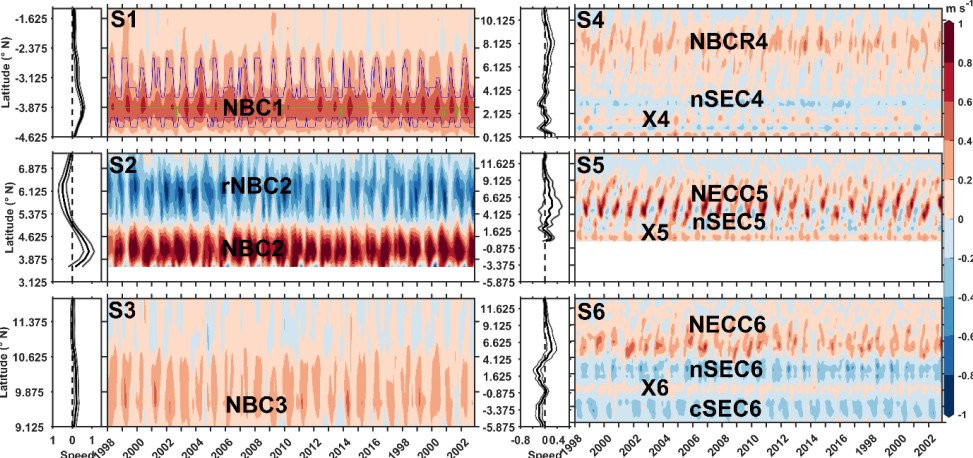

**Figure 3.** Hovmöller diagrams (1993 to 2017) of the cross-section current components (m s⁻¹) for S1, S2, S3 S4 S5 and S6. At the left of each diagram, the time series of time averages of the cross-section current (thick lines), framed by their corresponding standard deviations (thin lines) are plotted as a function of latitude. The red (blue) colors show the northward/eastward (southward/westward) directions of the cross-currents: North Brazil current (NBC), its retroflected branch (rNBC) and the retroflection signature at the limit between the NBC and the rNBC (NBCR); North Equatorial Countercurrent (NECC); the equatorial surface eastward flow named X; northern and the central branches of the South Equatorial Current (nSEC and cSEC, respectively). The numbers next to the acronym of the currents represents the number of the section. The green line in S1 indicates the time series of the maximum velocity of the cross-section current (NBC), and is framed by the time series of the maximum velocities divided by 2 (blue lines) over 1993-2017 period.





In order to further investigate the temporal variations of the current amplitude, the maximum
velocity values of each current vein identified in Table 1 have been extracted for each month
(corresponds to the current core and called Vmax hereafter: see Fig. 3, green line on S1). The
location of Vmax corresponds to the location of the current core. Then the corresponding
current width is estimated, considering the part of the section where the velocity is larger than
Vmax/2 (see Fig. 3, blue lines on S1). Finally, the relative intensity is computed by averaging
the cross-section velocity values over the estimated current width. For each current vein
crossing the six sections (Left column of Table 1), we compute the monthly time series of the
resulting Vmax, current width and relative current intensity/strength. Note that the latter has
also been computed by averaging velocity values over the full transect of each current vein
(e.g., not only where the velocity is larger than Vmax/2), but the results did not correctly reflect
the current variability observed in Fig. 3 and Fig. 4 (not shown). For the NBCR and the NECC,
the variability of the maximum velocity of two flow branches and their corresponding locations
is also analyzed, in order to compare with the study of Fonseca et al. (2004). It will also be used
to investigate the variability of the current location with respect to the wind variability and to
the tropical Atlantic climate modes. In this study, the presence of two branches of the e NBCR
and the NECC two-core structure are identified when the flow velocity profile shows two local
maxima separated by a local minimum respectively, in the NBCR region (considered between
4°-10° N) and the NECC region (considered between 3°-11° N) (Fig. 4S4-S6).






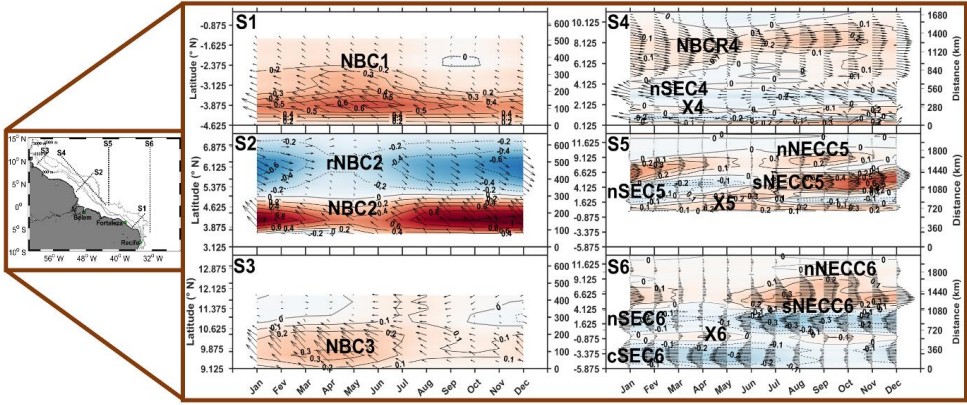

**Figure 4.** Hovmöller diagrams of the monthly climatology of the surface geostrophic (vectors superimposed on amplitude in m s⁻¹) over the 1993-2017 period. S1, S2, S3, S4, S5 and S6 are respectively for the sections 1 to 6 represented over the study area (Left map). nNECC and sNECC are respectively, the northern and southern branches of the North equatorial Countercurrent (NECC); NBC, the North Brazil Current; NBCR, the NBC retroflection limit; rNBC, the retroflected branch of the NBC, nSEC and cSEC, the northern and the central branches of the South Equatorial Current, respectively; and X, the equatorial surface eastward flow. On the right sides of each diagram, the distances from the lowest point (in km) are indicated.





## 4   Seasonal variability

Here we focus on the seasonal cycle of the different current branches observed in the study area. Therefore, a monthly climatology of the velocity estimates shown in Fig. 3 is calculated for each of the six sections (Fig. 4). For further analysis, the monthly climatology of Vmax, Vmax location, the current width and the relative current intensity has also been derived from the corresponding monthly time series for different current components ((Fig. 5).  Below, we analyze the results from the southeast to the northwest of the study area.





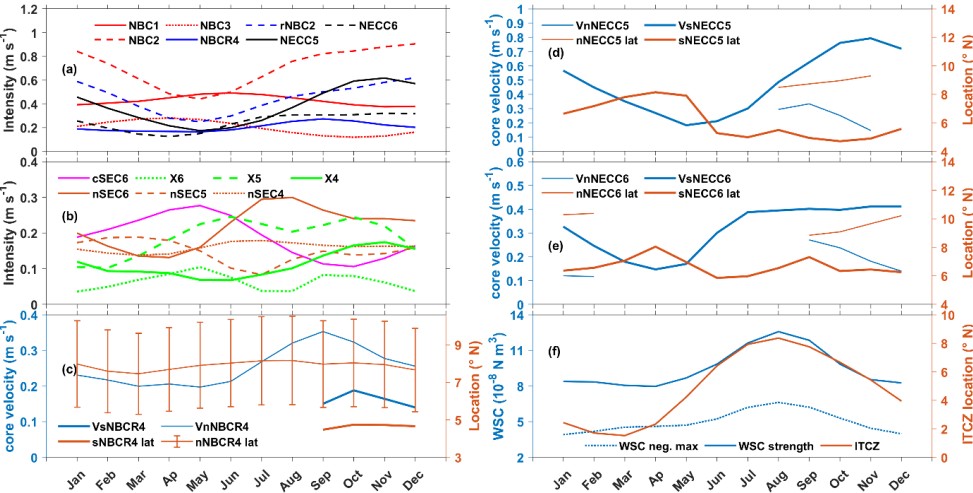

**Figure 5.** Monthly climatology of: the relative currents' intensity (m s$^{-1}$) **(a-b)**; the cores velocity/location in the North Brazil Current Retroflection (NBCR) regions **(c)**; the cores velocity/location of the North Equatorial Countercurrent (NECC) branches along sections 5 and 6 **(d-e)**; and the absolute values of the maximum negative wind stress curl (WSC neg. max), the WSC strength and the ITCZ location **(f)**. **(a)**: NBC1, NBC2 and NBC3 represents the North Brazil Current (NBC) on the crossing sections 1, 2 and 3, respectively; rNBC2 represents the retroflected NBC branch (rNBC) crossing section 2; NBCR4, represents the NBCR flow crossing section 4; NECC5 and NECC6 represent the NECC crossing 5 and 6, respectively. **(b)** nSEC4, nSEC5 and nSEC6 represent the northern branch of the South Equatorial Current (SEC) crossing sections 4, 5 and 6, respectively; cSEC6 represents the central branch of the SEC crossing section 6; and X4, X5 and X6 represent the equatorial surface eastward flow X crossing sections 4, 5 and 6, respectively. On **(c)**, **(d)** and **(e)**, the "V" initials/lat at the end of each acronym represent the core velocity/location (blue/orange color). The nNBCR/sNBCR and nNECC/sNECC represent the northern/southern core in the NBCR and the NECC regions, respectively.



### 4.1    The North Brazil Current and its retroflection

In Fig. 4, the NBC (Section 1) and the cSEC (Section 6) are observed both located at ~ 4° S and present similar seasonal cycles, with stronger flows during the first half of the year. The cSEC6 velocity maximum (0.3 m s$^{-1}$) / minimum (0.1 m s$^{-1}$) appears in May/October, respectively. The NBC1 velocity maximum (0.6 m s$^{-1}$) / minimum (0.4 m s$^{-1}$) appears in May-June/November-December (Fig. 5a). These annual cycles are also similar to the annual cycle of the NBUC transport observed by Rodrigues et al. (2007) at 10° S and which is related to the bifurcation location of the sSEC. This suggests that at its southernmost location, the seasonal variability of the NBC might be partly driven by the location of the sSEC bifurcation, which has been shown to be influenced by the annual cycle of the WSC over the area 5°–10° S, 25°–40° W (Rodrigues et al., 2007).

When comparing the NBC along sections S1, S2, S3 and S4 (Fig. 4S1-S4 and Fig. 5), it clearly depicts two different seasonal cycles along its northward path. The NBC2, NBCR4 and the rNBC2 flows show approximately the same seasonal cycles, in opposite phase with NBC1 and NBC3 ones, approximatively in phase. NBC2 is narrower but relatively stronger than NBC1. It decreases from January to May (by ~ 0.4 m s$^{-1}$), and then increases again to reach a maximum in November-December (1 m s$^{-1}$). Compared to NBC2, the rNBC2 is broader (width of ~ 350 km against 200 km) but less intense (maximum of 0.6 m s$^{-1}$ in November-December). In Fig. 4, from September to January, we observe two retroflections of the NBC: the main one around 8°N and a secondary one to the south, between 4-6° N (Fig. 4S4 and 5c). The flow of this secondary retroflection (called sNBCR) reaches its maximum intensity in October while the main retroflection flow (called nNBCR) reaches its maximum one month earlier. Then, it migrates northward to join the nNBCR4; and both are completely merged at the beginning of the following year (Fig. 4S4). Both branches merge at the beginning of the year and the NBCR then weakens to reach its minimum intensity in May. Note that the seasonal cycles of the NBC2 and the NBCR have are similar to the one of the NBC transport obtained by Johns et al. (1998) and Garzoli et al. (2004) using acoustic Doppler current profilers (ADCP)/Inverted Echos sounders/Pressure gauge data. Johns et al. (1998) related the seasonal cycle of the NBC observed in this area to the remote wind stress curl forcing across the tropical Atlantic. Here, we also see that the seasonal cycles of the NBC branches north of the equator (except the NBC continuity along the Guyana coast) follow the remote wind stress curl strength by one to four months (Fig. 5a-f). The northernmost location of the nNBCR maximum intensity occurs in August, when the maximum WSC strength is reached (Fig. 5c, f). The root mean square (rms)



of the monthly mean values of its location (Fig. 5c) is nearly constant (~2.3°), and is consistent
with the evidence that there may not be a preferred season for NBC ring formation (Garzoli et
al., 2003; Goni and Johns, 2003).
Further north, the NBC component flowing along the Guyana coast (NBC3) is twice wider (~
440 km) and less intense than the NBC2 (Fig. 4S2-S3). It reaches a minimum in October (~ 0.1
m s$^{-1}$), and a maximum during March-May (0.3 m s$^{-1}$) when the NBCR4 is minimum (Fig. 4S3-
S4 and 5). As already mentioned, its seasonal cycle is similar to the NBC1 and cSEC, and might
also be influenced by the sSEC bifurcation location.

### 4.2    The North Equatorial CounterCurrent

The NECC (both NECC5 and NECC6) seasonal cycle is similar to the ones of the NBC2,
NBCR4 and rNBC2 (Fig. 4S2, S4, S5-S6). It weakens along its pathway and its intensity is
maximum in November-December (~ 0.6 m s$^{-1}$ at 42° W and ~ 0.3 m s$^{-1}$ at 32° W), and
minimum in April-May (Fig. 5a). During the second part of the year, we observe the two-core
structure previously investigated by Urbano et al. (2006; 2008). The two cores/branches are
seen first at 42° W in August, then at 32° W in September (Fig. 4 and 5d-e). The northern
branch (nNECC) is narrower (located between 7°-9° N) and stronger (0.3 m s$^{-1}$) in August-
September at 42° W (Fig. 4S5 and 5d). It is even separated from the southern branch from
October to December. At 32° W, the NECC appears with one single core from September
(northern core velocity of ~ 0.2 m s$^{-1}$) to November. The northern core is gradually decreasing
in intensity and shifting northward until forming a separated second branch, which is located
between 9-11° N from December to February, and then becomes very weak in March-April
(Fig. 4S5 and 5e). From June to July, the NECC signature (sNECC branch) is located between
3-4° N at 42° W, and is connected from the south to the eastward flow associated with X5 in
Fig. 4S5.  Urbano et al. (2008) showed with ADCP data at 38° W that the eastward flow south
of the NECC corresponds to the mixing of NEUC and EUC waters during this period of the
year. The presence of the equatorial surface eastward flow X5 suggests that this latter may be
the one that favors the surfacing and the connection of both currents during June-July. From
June to November, the sNECC branch increases simultaneously with the rNBCR2. However, it
reaches its maximum one month earlier (November) and start decreasing when the rNBC is still
increasing. This confirms that the NECC is not only fed by the rNBC at the surface as suggested
by Verdy and Jochum, 2005. The sNECC witnesses the same variability at both 32° W and 42°
W, but reaches its minimum early in March-April at 32° W when the climatological NECC is
described in the literature as a reversing flow or is missing at its usual location (Garzoli and





Katz, 1983; Garzoli, 1992). At this location (32° W), the sNECC starts increasing from April-
May, between 4°-6° N far from the eastward flow X, and grows until November. Burmeister et
al. (2019) showed that, in the central Atlantic, when the ITCZ migrates northward (April to
August) (Fig. 5f), the nSEC recirculated eastward to reach the NEUC which then increases. The
presence of the sNECC flow in April-May may suggest that the NECC flow might be initiated
by the recirculation of the nSEC which should mix with the upper NEUC water during this
period. It reaches its first maximum in July-August together with the nSEC, and a second
maximum in November (Fig. 5a, b, e). At 32° W and 42° W the sNECC shows two northward
migrations. The first migration occurs from June-July (June) to August (September) at 42° W
(32° W), and the second occurs from October to April (Fig. 5d-e). Fonseca et al. (2004) also
found such behaviors but with some differences: two northernmost NECC locations in February
and August, and two southernmost NECC locations in June and December. But they lacked
data between March and May, and we do not use the same methods to compute the core
position.
**4.3    The central and northern branches of the South Equatorial Current**
Although the cSEC and nSEC are two branches of the westward SEC flow, they do not have
the same seasonal cycle (Fig. 4S4, S5, S6). The cycles of the nSEC4, nSEC5 and nSEC6 have
maxima at different periods of time. At 32° W, the nSEC (nSEC6) increases from April to reach
a maximum of ~ 0.3 m s$^{-1}$ in August, following the migration of the ITCZ (Fig. 5b, f). During
this time, at 42° W, the nSEC (nSEC5) migrates northward, and its intensity decreases until
July, when it almost disappears. The eastward flow X then appears (Fig. 4S5).  The nSEC5 is
observed again after July, increases and reach a maximum of ~ 0.2 m s$^{-1}$ in March (Fig. 4S5,
S6 and Fig. 5b). Over the continental shelf, most of the nSEC joins the NBC (section 4, around
2-4° N and 46° W) and a part of the flow deviates southeastward to join the rNBC and form the
eastward flow X4 (captured along section 4, Fig.4S4). The nSEC component that joins the NBC
reaches its maximum of ~ 0.2 m s$^{-1}$ between June and August (also following the ITCZ
northward migration, Fig. 5b, f). However, the nSEC seasonal variations are relatively small (~
0.15 m s$^{-1}$) (Fig. 4S4). It is particularly true for nSEC4 but the angle of the section relative to
the flow probably leads to a significant reduction of the current amplitude captured.
**4.4    The eastward current X**
Figure 4S4-S6 shows the presence of an eastward current near the equator. Such feature was
already observed and mentioned by Hisard and Hénin (1987) and Bourles et al. (1999b) using
hydrographic and ADCP data. In Fig. 4S4, eastward flows are captured between 1-2° N (X4)

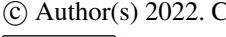



and along the equator. As mentioned in Sect. 4.3, they may be composed of the part of the nSEC
that does not join the NBC and the rNBC (which is kwown to feed the EUC in the thermocline
layer). The weaker intensity of X4, compared to X5 (i.e. X at 42) is explained by the angle
between S4 and the current direction. But the weaker intensity of X6 (i.e. X at 32° W) is due to
the weakening of the corresponding eastward flow between both longitudes. X4 and X5 are
observed almost throughout the year (Fig. 4S4-S6) and their amplitude follows a semi-annual
cycle (Fig. 5b) similar to the EUC in the eastern Atlantic (Hormann et al. 2007). However, the
periods of the maxima are slightly different from location to another. X4 displays a semi-annual
cycle, with a weak maximum in March-April ($\sim 0.1$ m s$^{-1}$) and another maximum in November
($\sim 0.2$ m s$^{-1}$). X5 amplitude is larger because of the merging of X4 with the eastward flow seen
along the equator (Fig. 4S5). Its maxima occur in June and October with similar intensity (more
than 0.2 m s$^{-1}$) (Fig. 5b). X6 (at 32° W) is weaker, but reaches its maxima in May and
September-October (less than 0.1 m s$^{-1}$). Since this eastward X flow is almost not documented
in the literature, we will come back to it in Sect. 6.
**5   Interannual variability**
Beyond the dominant seasonal variability of the circulation at regional scale, we also observe a
year-to-year variability of the surface velocities in the study area (Fig. 2 and Fig. 3). The latter
is analyzed here, using the time series of the characteristics of the different current branches
(see end of Sect. 3) captured along the 6 sections (Fig. 6). We will also analyze this variability
in the light of the tropical Atlantic climate modes.



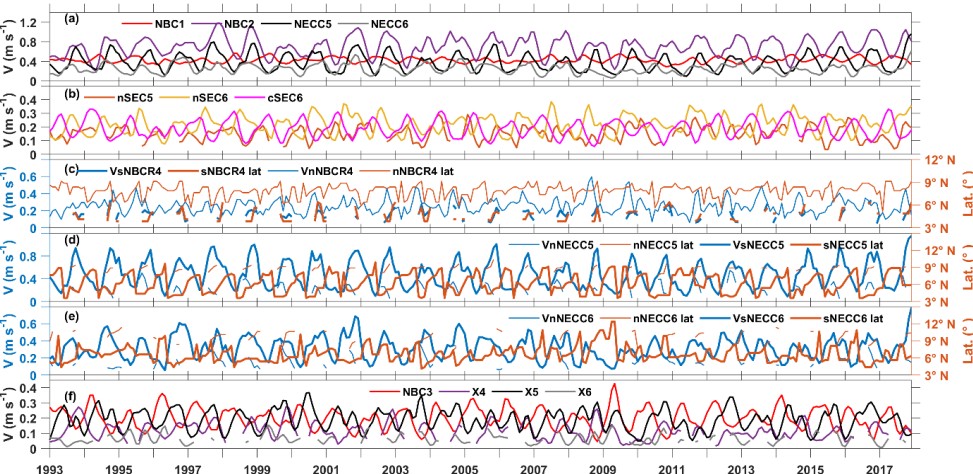


**Figure 6.** Time series of the 4-month low-pass filtered characteristics of the geostrophic currents **(a-f)** captured
along sections 1 to 6. **(a)**: intensity of NBC crossing the sections 1 (NBC1) and 2 (NBC2), and of the NECC
crossing the sections 5 (NECC5) and 6 (NECC6). **(b)**: intensity of the cSEC crossing the section 6 (cSEC6) and
of the nSEC crossing the sections 5 (nSEC6) and 6 (nSEC6). **(c)**, **(d)** and **(e)** core velocities and the locations of
the northern and southern cores of respectively the NBCR flows crossing the section 4 (NBCR4), and of the NECC
cores on sections 5 (NECC5) and 6 (NECC6). **(f)** NBC crossing the section 3 towards the Guyana coast (NBC3)
and the equatorial surface eastward flow crossing the sections 4 (X4), 5 (X5) and 6 (X6). The "V" initials at the
beginning, and "lat" at the end of each acronym of the legends in **(c)**, **(d)** and **(e)** represent the core velocity and
location, respectively (blue and orange colors, respectively). The thin (tick) lines represent the northern (southern)
branches of these currents/flows.





In Fig. 6a-b we observe that over the whole study period the intensities of the cSEC6 (cSEC
along section 6) and of the NBC1 (NBC along section 1) and NBC2 (NBC along section 2)
vary between 0.05-0.35 m s$^{-1}$, 0.3-0.6 m s$^{-1}$ and 0.2-1.2 m s$^{-1}$, respectively. with corresponding
mean values of 0.2 m s$^{-1}$ ± 0.06, 0.4 m s$^{-1}$ ± 0.06 and 0.7 m s$^{-1}$ ± 0.2. To the north, when crossing
section 3, the NBC weakens and ranges between 0.05-0.43 m s$^{-1}$ with a mean intensity of 0.2
m s$^{-1}$ ± 0.07. In the equatorial region (±5° of latitude), along sections 5/6, the nSEC intensity
varies between 0.05-0.3 m s$^{-1}$/0.1-0.4 m s$^{-1}$, with a mean value of 0.15 m s$^{-1}$ ± 0.05/0.2 m s$^{-1}$ ±
0.07 (Fig. 6b). The eastward flow X intensity vary between 0.15-0.3 m s$^{-1}$, 0.05-0.4 m s$^{-1}$ and
0-0.15 m s$^{-1}$ when crossing sections 4, 5 and 6, respectively, with corresponding mean values
of 0.12 m s$^{-1}$ ± 0.05, 0.2 m s$^{-1}$ ± 0.07, and 0.06 m s$^{-1}$ ± 0.03.
As mentioned in Sect. 3, the interannual variability is more important in the eastern part of the
basin, and especially in the NECC region. The two-core structure of the NECC and the NBCR
regions show the highest year-to-year variations in both velocity and location (Fig. 6c-e). The
NECC/NBCR time series were found significantly correlated (>0.98) in terms of both intensity
and core velocity (sNECC and nNBCR; Figure not shown). At 42° W, the sNECC/nNECC)
velocity core varies between 0.1-1.2 m s$^{-1}$/0.05-0.55 m s$^{-1}$, with a mean value of 0.5 m s$^{-1}$ ± 0.25
/0.25 m s$^{-1}$ ± 0.1. It is located between 3.6°-9.9° N/7.6°-10.4° N, with a mean location at 6.1°
N ± 1.7°/8.9° N ± 0.8° (Fig. 6d). At 32° W, it varies between 0.05-0.8 m s$^{-1}$/0.05-0.5 m s$^{-1}$, with
a mean velocity of 0.3 m s$^{-1}$ ± 0.13/0.2 m s$^{-1}$ ± 0.09. It is then located between 4.1°-12.4° N/7.4°-
11.6° N, with a mean location of 6.6° N ± 1.4°/9.7° N ± 1.1° (Fig. 7e). The sNBCR /nNBCR
maximum velocity crossing the section 4 varies between 0.05-0.3 m s$^{-1}$/0.05-0.6 m s$^{-1}$, with a
mean value of ~ 0.15 m s$^{-1}$ ± 0.06/0.25 m s$^{-1}$ ± 0.08. It is located between 3.9°-6.4° N / 5.1°-
9.1° N, with a mean location of 4.7° N ± 0.8°/7.9° N ± 0.8°. Northeast of the equator, for all
these current branches, we observe important year-to-year variations, both in terms of current
core location and of velocity amplitude.
For further analysis, the anomalies of the current's characteristics (intensity and core
value/location) relative to their monthly climatology have been computed (Fig. 7). First, we do
not see obvious relationship between the different resulting monthly anomaly time series over
the whole period investigated. No relationship was found between the evolution of the NECC
branches intensity/core value and their location, or the NBCR flows maximum velocity and
their location. However, for some particular years, both the NECC intensity/core velocity and
location show significant anomalies at 32° W. For example, the monthly anomalies of the
sNECC6 location were shifted far to the north / south in 2009 and 2010 / 1996 and 2001.





Simultaneously, the monthly anomalies of the sNECC6 intensity/core value were unusually
weak / strong (Fig. 7f-g).
To investigate the relationship between the AMM, the AZM, and the characteristics of the
different current branches between 1993 and 2007, we computed the three-month running mean
of both the monthly anomaly time series and of the climate mode indexes (so-called 3-month
anomaly time series). The latter have been correlated with each other in order to learn more
about the spatio-temporal characteristics of the interannual variability in the study area (Figures
not shown). Only the correlations greater than ±0.5 and which have been found significant with
95 % of confidence level, performing the Student test are discussed below (listed in Table 2).



















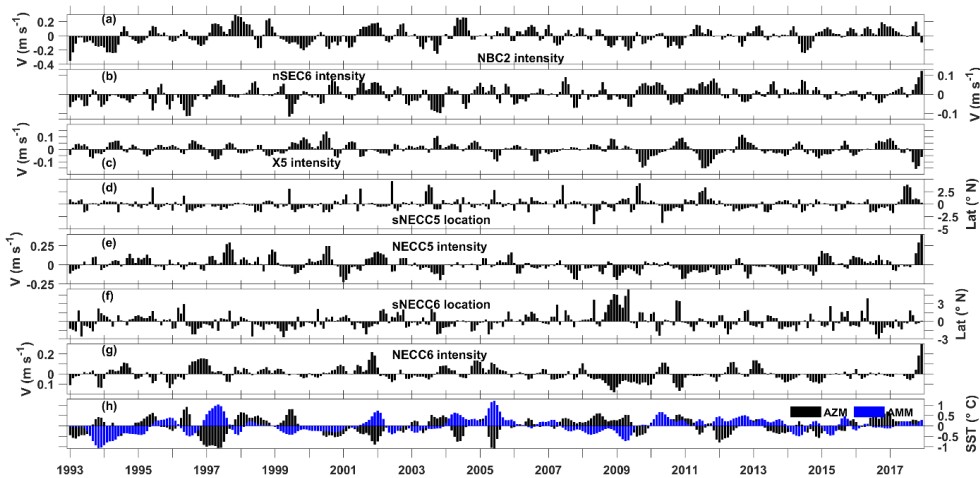

**Figure 7.** Time series of monthly anomalies relative to the monthly climatology for: **(a)**, the NBC intensity on the section 2; **(b)**, the nSEC intensity on the section 6; **(c)**, the equatorial surface eastward flow X intensity on the section 5; **(d)**, the sNECC core location on the section 5; **(e)**, the NECC intensity on the section 5; **(f)**, the sNECC core position on the section 6; and **(g)**, the NECC intensity on the section 6; **(h),** Normalized indexes of the tropical Atlantic meridional mode (AMM: blue color) and zonal mode (AZM; black color).



**Table 2.** Correlations values between the characteristics of the current branches and the Atlantic meridional and
zonal mode indexes (AMM and AZM, respectively). Where the correlations are found lower than 0.5, we indicate
"insignificant" over the whole time period ("none"). The current branches analysed are the NBC (North Brazil
Current), sNECC (southern branch of the NECC), nSEC (the northern branch of the South Equatorial Current) and
the equatorial eastward flow called X.

| Current branches' characteristics | | Atlantic meridional mode (AMM) during March-April-May | Atlantic Zonal mode (AZM) during June-July-August |
|---|---|---|---|
| NBC1 intensity (section 1) | Correlation | Higher than -0.51 | Insignificant |
| | Period | June-July | None |
| NBC2 intensity (section 1) | Correlation | 0.58 | Insignificant |
| | Period | March-April-May | None |
| sNECC5 intensity (section 5) | Correlation | Insignificant | -0.51 |
| | Period | None | September |
| sNECC6 intensity (section 6) | Correlation | Higher than 0.50 | Insignificant |
| | Period | March-April-May | None |
| sNECC6 core location (section 6) | Correlation | Higher than 0.51 | Insignificant |
| | Period | March-April-May | None |
| nNECC6 core location (section 6) | Correlation | -0.62 | Insignificant |
| | Period | March | None |
| nSEC5 intensity (section 5) | Correlation | Insignificant | -0.52 |
| | Period | None | November |
| nSEC6 intensity (section 6) | Correlation | Higher than 0.52 | Insignificant |
| | Period | March-April-May | None |
| Equatorial eastward flow X (Sections 4 and 5) | Correlation | Higher than -0.55 | Insignificant |
| | Period | May-June | None |
| Equatorial eastward flow X (Section 6) | Correlation | Higher than 0.62 | Higher than -0.52 |
| | Period | March-April-May | June-July |












The AMM index is found anticorrelated with the NBC intensity along the section 1 during
March-April-May, the nNECC core location along the section 6 and the equatorial eastward
flow (X) intensity west of 42° W, with respective coefficient of correlation (cc) higher than -
0.51 during June-July, about -0.62 in March, and higher than -0.55 in May-June. These anti-
correlations show the probability of the positive (negative) AMM phases to lead the negative
(positive) anomalies of the nNECC core location at 32° W with 0-month delay, the negative
(positive) NBC intensity between 3°-5° S and the X flow intensity anomalies (west of 42° W)
with 1- to 2-month and 0- to 1-month delay, respectively. However, the AMM index during
March-April-May is found correlated with the NBC intensity north of the equator before the
retroflection (cc ~0.58) and, the sNECC intensity/core location and the nSEC intensity at 32°
W (cc higer than 0.50/0.51 and cc higher than 0.52, respectively) during the same period of
time. This suggests that the positive (negative) AMM phases probably drives the positive
(negative) anomalies of the corresponding currents with no time lag.
During June-July-August, the AZM index is found anticorrelated with the sNECC/nSEC
intensity at 42° W (cc=-0.51/-0.52) only in September/November. It suggests that the positive
(negative) AZM phases probably lead the negative (positive) anomalies of the sNECC/nSEC
intensity with 1- and 3-month delay, respectively.
The eastward flow X intensity at 32° W is found simultaneously correlated with AZM and
AMM in June-July and March-April-May, with cc higher than 0.62 and -0.52, respectively.
This suggests that the positive (negative) anomalies of X at 32° W might be associated with
both positive (negative) AZM and negative (positive) AMM phases with no delay.
Referring to Cabos et al. (2019), the relationships found between the currents and the AMM
show the influence of the strengthening / weakening of the southeast trade winds in the southern
hemisphere on the southward / northward migration of the nNECC core at 32° W, whereas the
NBC intensity between 3°-5° S and the equatorial eastward flow X intensity west of 42° W
decrease / increase. Conversely, the strengthening / weakening of the southeast trade winds in
the southern hemisphere should influence the northward / southward migration of the sNECC
core at 32° W, whereas the NBC2, sNECC6 and nSEC6 intensities increase / decrease.
Referring to the same authors, the relationship with the AZM should indicate the probable
influence of the positive / negative westerlies anomalies in the western part of the basin on the
negative / positive anomalies of the sNECC and nSEC intensities at 42° W. Concerning the
eastward flow X at 32° W the relationship with both the AZM and the AMM modes should
indicate its strengthening / weakening during the concurrent events of positive / negative



westerlies anomalies in the western part of the basin and the negative / positive southerlies
winds anomalies in the southern hemisphere.

## 6    Discussion

Finally, we have computed the seasonal maps of the geostrophic currents in the whole WTA
(Fig. 8) in order to have a regional view of the seasonal variations of the circulation. Fig. 8
confirms the results obtained from the analysis of the cross-section velocities in Sect. 4 (in terms
of seasonal cycles and spatial structure) but also highlights interesting new features.
A large cyclonic circulation is observed between 35°-45° W and 0°-5° N during boreal spring
(blue ellipse). The latter is formed by the westward nSEC which is suddenly deviated to the
northeast by the presence of an eastward flow at ~32° W. Then, near 44° W and 5° N, the nSEC
meets the rNBC which reaches it southernmost position during this season, and deviates to the
southeast. When reaching the equatorial region between 0°-2° N, where the equatorial eastward
flow is found (Fig. 4), the resulting flow becomes stronger, and is deviated to the east, close
this cyclonic feature. This finding might answer the question of Schott et al. (1998) about the
destination of the rNBC during spring, when the rNBC does not feed the NECC anymore.
During the boreal winter, another cyclonic circulation is observed between 44°-50° W and 5°-
10° N (blue ellipse in Fig. 8): part of the NECC recirculates north-westward to join the rNBC.
During both boreal winter and spring, a south-westward recirculation of the NECC appear to
strengthen the nSEC located west of 32° W. It is consistent with the increase of the nSEC
intensity observed along section 4, compared to the nSEC intensity captured along section 6,
between February and May (Fig. 5b).
During the second half of the year, Fig. 8 shows a wider NECC which extends north of 10°N.
During boreal fall, the NECC flow is formed by a nNECC branch separated from the initial
sNECC branch during between 38°-48° W, that meet east of 38° W. This is consistent with Fig.
4 S5-S6. During boreal summer, the nNECC branch seems to be supplied by the northern part
of the NBC retroflection. This connection seems to fade during boreal fall.
In the equatorial region (2° S-2° N), Fig. 8 also shows an equatorial eastward flow X with lower
amplitude which appears to be extended east of 32° W and stronger during boreal spring and
fall. This feature can be related to the near-surface eastward flow mentioned by Hisard and
Hénin (1987) and Bourlès et al. (1999b) on top of the EUC in the WTA. Hisard and Hénin
(1987) explained the poor description of this current in the literature by the difficulty of ADCPs



measurements to fully capture the upper layer currents in this area. They showed that this near-
surface current, independent from the EUC, can reach amplitudes larger than 0.5 m s$^{-1}$ between
23° W and 28° W. In order to understand this difference in current amplitude, compared to ours
(~ 0.1 m s$^{-1}$), we computed the seasonal maps of the near-surface Ekman currents (Fig. 9). We
observe westward currents with higher amplitudes in this equatorial band (larger than 0.2 m s$^{-1}$
). Then, we conclude that the near-surface velocities found in this study in the equatorial region
are underestimated, particularly in the eastern basin, compared to the absolute surface velocities
(Fig. 5b).
For the first time, the seasonal cycle of the equatorial eastward flow has been analyzed in Sect.
4.4 of this study (Fig. 4S4, S5, S6 and Fig. 5b). It is similar with the seasonal cycle of the EUC
(semi-annual cycle with two maximum), which might be due to the fact that most part of the
flow is fed by the rNBC.










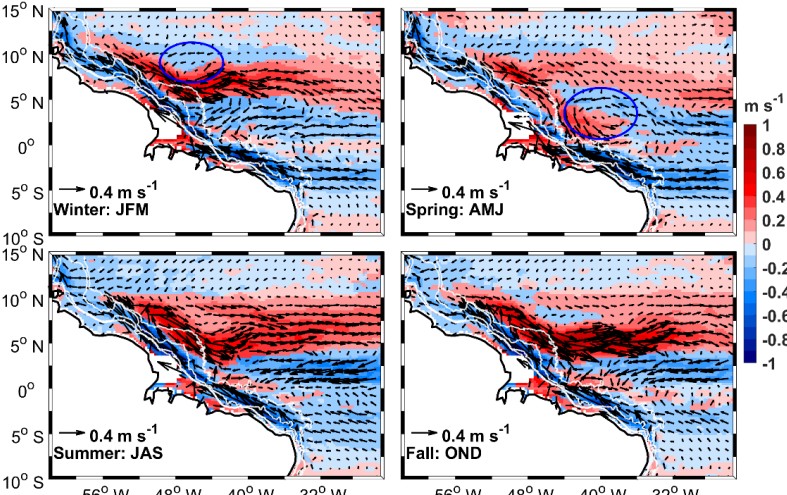

**Figure 8.** Seasonal maps of the geostrophic currents in the western tropical Atlantic over the 1993-2017 period for boreal winter (top left panel: JFM for January-February-March), boreal spring (top right panel: AMJ for April-May-June), boreal summer (bottom left panel: JAS for July-August-September:) and boreal fall (bottom right panel: OND for October-November-December:). The velocity vectors are superimposed on the velocity amplitudes multiplied by the sign of their horizontal components (m s$^{-1}$). The two cyclonic circulations observed during boreal winter and spring are indicated by blue ellipses. The white lines near to the continent are from west to east, the 300 m, 1000 m, 3000 m and 4000 m isobaths.



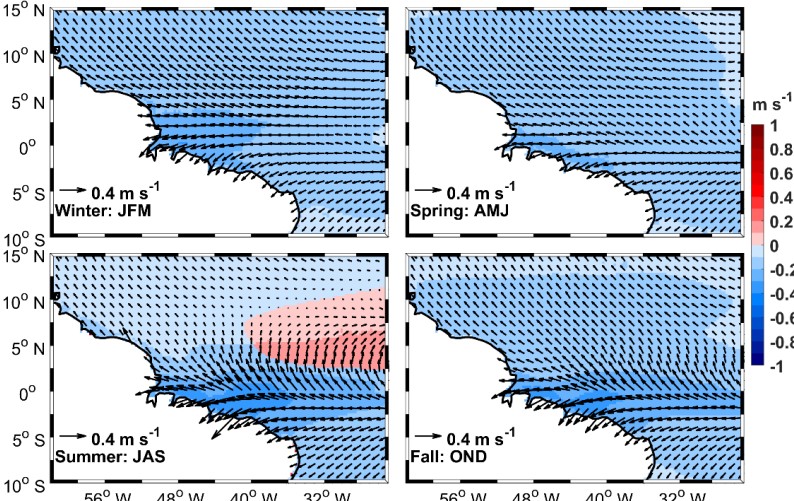

**Figure 9.** Seasonal maps of the Ekman currents in the western tropical Atlantic over the 1993-2017 period for boreal winter (top left panel: JFM for January-February-March:), boreal spring (top right panel: AMJ for April-May-June), boreal summer (bottom left panel: JAS for July-August-September) and boreal fall (bottom right panel: OND for October-November-December). The velocity vectors are superimposed on the velocity amplitudes multiplied by the sign of their horizontal components (m s$^{-1}$).



Finally, the current analysis made above, allows us to derive a new scheme of the seasonal
variations of the western boundary tropical Atlantic circulation (Fig. 10). The new current
branches found in this study are indicated in green and the currents coming from the north/south
are in blue/orange. The width of the current arrows is proportional to the current amplitude and
the dotted arrows represent the currents with the minimum amplitudes. C1 and C2 are for
respectively, the cyclonic circulations highlighted between 44°-50° W and 35°-45° W.















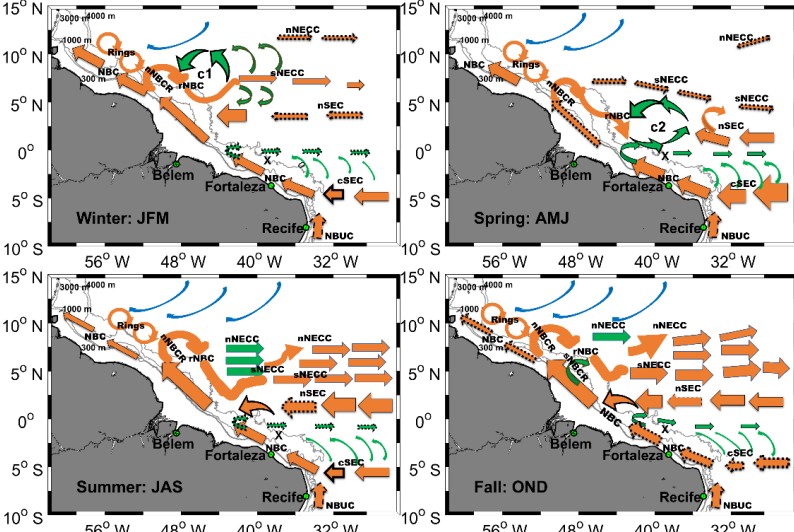


**Figure 10.** Schematic view of the seasonal maps of the tropical western boundary circulation. C1 and C2 represent
the cyclonic circulations highlighted in this study. The wider (dotted thin) arrows show current branches with
maximum (minimum) intensity. The subsurface NBUC is represented by dashed arrows. The already known
current branches are in orange/blue. The green arrows characterize the new branches observed. NBC is the North
Brazil Current; nNBCR and sNBCR are the northern and the southern flows of the North Brazil Current
retroflection, respectively; rNBC is the retroflected branch of North Brazil current; nNECC and sNECC are the
northern and the southern branches of the North Equatorial Countercurrent (NECC), respectively; cSEC and nSEC
are the central and the northern branches of the South Equatorial Current (SEC), respectively.














Concerning the interannual variability, Hormann et al. (2012), based on the analyses of 17 year
of altimetry and drifter' data, found interesting scenarios about the NECC spatial and temporal
variability. However, they did not separate the sNECC from the nNECC. In their study, the
authors have associated the strengthening of the NECC in the whole basin to the negative phase
of the AZM, and its northward shift to the positive phase of the AMM. Our results show
different behaviors of the NECC system as a function of the branch and core location. We also
showed possible relationships between the southern NECC branch intensity and location with
the AMM phases at 42° W, and conversely a possible relationship between the southern NECC
branch intensity and location with the AZM phases at 32° W. These results open the way to
deeper investigations in future studies.
**7    Summary and Perspectives**
Twenty-five years (1993-2017) of gridded altimetry data from CMEMS were used to improve
the description of the seasonal and interannual variations of the western boundary circulation
of the tropical Atlantic. Therefore, we defined six horizontal sections crossing the main upper
layer currents' veins. C3S ERA5 wind estimates and the NOAA OI SST v2 product were also
used to investigate the possible link between the variability of the regional circulation and the
large-scale remote wind forcing on one hand, and the tropical Atlantic climate modes on the
other hand.
Our results highlight a complex regional circulation, with significant seasonal and year-to-year
variations of the currents' intensity and location. South of the equator, we observe a
stronger/weaker central branch of the South Equatorial Atlantic (cSEC) and of the North Brazil
Current (NBC) during boreal spring/fall. North of the equator, the NBC component flowing
along the Guyana coast exhibits a similar annual cycle. However, between both, the NBC part
located before the retroflection is out of phase (i.e. stronger during summer-fall), when fed by
the northern branch of the South Equatorial Current (nSEC). Its larger amplitudes appear during
boreal fall, 3-4 months after the maximum of the remote wind stress curl (WSC) strength (in
August). The North Equatorial Countercurrent (NECC) is connected with the retroflected
branch of the NBC, and both show similar annual cycle. A secondary North Brazil Current
Retroflection (NBCR) was observed for the first time during boreal fall in this study: it is
located between 4°-6° N. The two-core/branch structure of the NECC during the second half of
the year was also confirmed and analyzed separately. Between 0°-5° N and 35°-45° W, a surface
cyclonic circulation develops during boreal spring. It is found to initiate the growth of the





NECC at 42° W in June. However, at 32° W, the NECC doesn't show any connection with this
cyclonic circulation, but starts its seasonal cycle earlier in April when the ITCZ migrates
northward and the remote WSC strengthens. In the equatorial region, between 2° S-2° N, the
geostrophic currents show the presence of an equatorial surface eastward flow which has a
seasonal cycle similar to the Equatorial Undercurrent (EUC). But he near-surface Ekman
currents have to be taken into account here to have a good description of the surface circulation.
Concerning the interannual variability, it is much weaker than the seasonal one. We do not
observe an obvious picture at regional scale but it is more important in the eastern part of the
basin. The analysis of the changes in characteristics of the different current branches (intensity
and core location), with respect to the tropical Atlantic climate modes, shows different possible
scenarios associated with one or both modes. It opens the way to further investigations
concerning the link between the Atlantic climate modes (ACM) and the current transports, not
possible with altimetry alone.
As a conclusion, this study demonstrates the ability of altimetry to characterize the seasonal
and interannual variability of the surface circulation in the study area. It confirm previous
findings but also significantly complements the knowledge of the different currents at regional
scale. Combined use of regional modelling, altimetry and in-situ observations will allow to go
further in the understanding of the spatial and temporal structure of the regional circulation.
The intraseasonal variability, significant in the near-shore region of the study area (Fig. 2d) is
not studied here. It will be the subject of a future work, based on a coastal altimetry product
that will allow a significantly better resolution and accuracy along the continental shelf,
compared to a gridded product.
**Authors contribution**
Djoirka M. Dimoune performed the data analyses as part of his PhD thesis. Florence Birol,
Fabrice Hernandez, Fabien Leger and Moacyr Araujo supervised this research.
**Competing interests**
The authors declare that they have no conflict of interest.



**Acknowledgements**
This work has been supported by: CAPES Foundation who funded the thesis of the first author,
CAPES-Print (grants 88887.467360/2019-00) who funded his visit to LEGOS during the first
six months spent in the laboratory, and LEGOS and the CNES/TOSCA program as part of
SWOT-Brésil project that funded the three last months of extension in the laboratory. The
authors thank CMEMS, who processed the geostrophic currents data and made them available.
They thank also ECMWF and NOAA respectively who made available the mean wind fields
data and SST for the work. Thanks also to Joël Sudre from LEGOS who developed the
GEKCO product used for this study. This work represents collaboration by the INCT
AmbTropic, the Brazilian National Institute of Science and Technology for Tropical Marine
Environments, CNPq/FAPESB (grants 565054/2010-4 and 8936/2011 and 465634/2014-1) and
the Brazilian Research Network on Global Climate Change FINEP/Rede CLIMA (grants
01.13.0353-00). This is also a contribution to the LMI-TAPIOCA and to the TRIATLAS
project, which has received funding from the European Union's Horizon 2020 research and
innovation program under grant agreement No 817578

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
