# Peer review of "Revisiting the tropical Atlantic western boundary circulation"

_EGUsphere, 2022_

## Referee Comment (RC1)

**Review of Dimoune et al.: "Revisiting the tropical Atlantic western boundary circulation from a 25-year time series of satellite altimetry data"**

This study uses 25 years of satellite altimetry data to describe the mean surface circulation of the Western Tropical Atlantic. The authors describe in detail the seasonal cycle of different branches of the South Equatorial Current, the North Brazil Current and the Guyana Current at different zonal and meridional transects. A main novel result of this manuscript is the description of a current branch at 0-2°N, above the Equatorial Undercurrent. This surface branch was previously unremarked in the literature and appears to be an extension of the North Brazil Current retroflection. Consistent with the literature, the results show that some of these branches have in-phase seasonal cycles, peaking in late winter/early spring, whereas others peak in fall. The interannual variability of the circulation is related to the Tropical Atlantic Meridional Mode.

This is study is a relevant update of the mean and low-frequency variability of the surface circulation in a globally important region of the tropical ocean, where vigorous inter-hemispheric exchanges take place. The analysis is simple but robust and the manuscript is organized logically. I have to admit, however, that I had a hard time getting through the text, which is terse and acronym-laden. Besides addressing the technical points below, I strongly recommend the authors work on their text to make it accessible to those who do not work on the Tropical Atlantic oceanography every day.

**Specific and technical points**

1. **Geostrophic currents near the Equator**
   The manuscripts lacks a description of the robustness of the equatorial $\beta-$plane approximation for calculating geostrophic velocities at $\pm 3°$ of the Equator. How accurate are these velocities? Can you really trust small changes in speed across the Equator (e.g., described in section 3)?

2. **Intraseasonal variability**
   From figure 2, the intraseasonal/subsesonal variability ($< 120$ days) is as important as the seasonal variability. The authors should discuss the intraseasonal variability instead of the interannual variability.

3. **Some methodological details are missing**
   The authors should improve their description of their estimation of various current properties. For example, how do the authors estimate thee properties in table 1? Are the authors simply choosing are maximum velocity in the transect? How about the

current width? Are the authors eye-balling this property from year transect? Wouldn't fitting a functions (e.g., a gaussian) to the cross-track velocity profile be a more effective way to estimate those properties? Also, does the interpolation of satellite altimetry sea-level from along-track to regular grid smear the currents? What's the effect of this interpolation on the width and intensity of the currents? Please, discuss.

4. **Figures are diddicult to read**
   Most figures are barely readable. The labels are tiny and oftentimes there are too many lines in the plots. The authors should increase the labels, rearrange the panels, and try to use less lines to improve readability.

5. **Tidal correction on the Amazon Shelf**
   I think the "erroneous altimetry measurements" mentined around line 160 is actually poor tidal corrections on the Amazon Shelf. The authors should more clearly described what they consider unrealistic values. What critereon are the authors using to remove those values?

6. **Wind-driven currents**
   As the name suggests, the GEKCO product contains both geostrophic and Ekman currents. If the authors are interested only in Ekman currents for figure 9, why don't the authors calculate those directly from ERA5?

**Typing, English and minor technical corrections**

1. line 43: change works with studies.

2. line 51: (...) North Brazil Under Currents (NBUC) which raises to the surface around → (...) North Brazil Under Currents (NBUC), which surfaces around.

3. line 77: meaning of "no more respected" is unclear. Do you mean "no longer satisfied"?

4. line 92: Do you mean understudied?

5. lines 108, 110 and elsewhere: "to allow" is a transitive verb—something or somebody allows somebody to do something else. So the dataset allows you to provide a more robust (...).

6. line 111: as follow → as follows.

7. line 125: Why the hat in Goês?

8. line 158: averaged on a monthly basis → averaged monthly.

9. line 183 and elsewhere: Pound → Pond. (Also, those should be 10-m winds, right?)

10. line 184/eq. (1) and (2): plese use $\tau$, not $\zeta$, to refer to wind stress. Also, $x$ and $y$ in eq. (3) should in subscript: $\tau_x$ and $\tau_y$.

11. lines 196-197: why use zonally average wind instead of local winds?

12. line 218: why is Guyana in figure 2?

13. line 221 and elsewhere: Do you mean path instead of vein?

14. line 261: more than → longer than.

15. line 261: less than → shorter than.

16. Figure 2: you should mention in the caption that the colorbar of (c) is different than the ones of (b) and (d).

17. line 291 and elsewhere: **to name a current or current branch X** is a horrible idea. Please, be creative and come up with a more descriptive name.

18. line 393: lowest → southernmost.

19. line 413: remove extra parenthesis.

20. line 428: currents intensity → speed.

21. line 484: Further north → Farther north.

22. line 517: recirculate → recirculates.

23. line 591-592 and elsewhere: what does ± stands for here? Standard derivation? Standard error?

24. line 630: Student test → Student's test.

25. line 670: analysed → analyzed.

26. line 743: ADCPs measurements → ADCP measurements.

27. line 753: similar with → similar to.

28. line 870: he → the.

29. line 880: confirm → confirms.

---

## Referee Comment (RC2)

Review of "Revisiting the tropical Atlantic western boundary circulation from a 25-year time series of satellite altimetry data" by Djoirka M. Dimoune, Florence Birol, Fabrice Hernandez, Fabien Léger, Moacyr Araujo [Paper # egusphere-2022-402]

The study uses a gridded product of geostrophic surface velocities based on satellite altimeter observations to determine the seasonal and interannual variability of the surface circulation in the western tropical Atlantic. In addition to this surface velocity product, Ekman currents are used from another gridded product (GEKCO) as well as gridded surface winds from ERA5 in order to relate the geostrophic surface velocity variability to changes in the wind field. Furthermore, a sea surface temperature product is used to investigate the relation of the surface velocity variability to variability associated with the Atlantic climate modes, the meridional as well as the zonal mode.
In order to investigate the seasonal and interannual variability of the individual current branches, 6 sections are defined in order to determine the strength and position of the individual current branches and to compare their variability patterns among each other.

The results show that a large fraction of the currents show a similar seasonal cycle, while a second regime seems to exist showing a somewhat different seasonal evolution, but agrees amongst each other.
An eastward current within the equatorial region and west of 42°W is identified and related to a larger scale cyclonic circulation.
The NECC is found to have a double core structure, which is somehow related to seasonal chances in the wind stress curl.
Interannual variations of the NBC seem to be opposite in the different hemispheres, which are related to the Atlantic meridional mode phases as well as the interannual variability of the NECC and nSEC at 42°W. Interannual variability of the NECC and nSEC at 32°W is related to the zonal mode phases.

Formal review:
The manuscript is well written and organized. Understanding the seasonal to interannual variability of the western tropical Atlantic surface circulation and its forcing mechanism is essential in order to understand the transport of heat, salt and other tracers across the equator and hence between the hemispheres and therefore definitely deserves attention. The methods used to investigate the seasonal to interannual variability are well explained, however I miss a more sophisticated method to corroborate the relation of the assessed variability to the forcing mechanisms in terms of the variability in the wind field and the different phases of the Atlantic climate modes. This study cites Hormann et al. 2012, where a lot of effort is put into the investigation of this relation e.g. regression and composite analysis etc. Maybe something like this could be included here. In my opinion as gridded products are used here only and hence no need for own calibration or gridding methods,  more effort should be put in the validation of these products with other available data sets.
In total I find the subject of the study important, but I recommend that some additional methods should be included in order to corroborate the findings of the study.

General remarks:

As mentioned before only gridded products are used here, which are readily available. However, these products based on geostrophy and Ekman all need to be adapted within the equatorial region as geostrophy and Ekman do not hold there due to the vanishing Coriolis parameter. I think more effort and/or explanation have to be included to convince the reader that these products are trustworthy in this region e.g. maybe compare the equatorial velocities to surface velocities measured at the PIRATA buoy sites? Along these lines which data set can be used for comparison of the Amazon outflow? It is just stated that the values of the surface velocity product there are unrealistic, but it needs some sort of reasoning to say so, before just blanking out these values.

Another point about the products: An Ekman velocity product is used, which is available and then the wind stress is calculated from the ERA5 winds. However, calculating the wind stress relies on empirical functions as explained in Line 184 and hence is crucially dependent on the formula used. In my opinion this means using an Ekman product on the one hand and calculating wind stress from some wind product on the other hand could lead to wind stress and Ekman product not necessarily fitting together, which is maybe not so nice when interpreting the results or at least this should be clearly stated somewhere. Are e.g. the empirical functions for wind stress to determine the Ekman products the same or do they differ?

To me it gets not exactly clear what are the new results from this study. For some of the results it is stated that it agrees with former studies and I might also miss some literature information, but you should definitely make point out more clearly what are the new findings here.

The detection of current X seems a somewhat new result as I understand. However, it is based on velocities from the equatorial band, which are critical for the products in use in this region (see my comment above). Hence, I think it is crucial at this point to validate the products in the equatorial region in order to make clear that this current X is not an artefact of the method to obtain the velocities in this region.

In addition to these general remarks, I have some detailed remarks throughout the text:

Detailed remarks:

Line 39-41: What do you mean here? Return branch of the AMOC influenced by the wind? Maybe use the statement of Schott et al. 2005 here? "The surface circulation in this region is a superposition of the AMOC, the flow related to the Subtropical Cells and Sverdrup dynamics." In general, I miss that the STCs are mentioned here. They are shallow overturning cells connecting the tropics and subtropics and also have their imprint on the surface circulation.

Line 67-70 What about the formation of the NBC rings? This is also quite characteristic of the surface circulation and should be mentioned here.

Line 72: Which means in general Sverdrup dynamics apply here (why do you separate into Ekman and geostrophic velocities here?) superimposed by the AMOC and STC (see my comment above), even if west of 32°W this does not apply anymore as you mention later.

Line 86: response **to** the the wind stress curl

Line 92: So then should your study maybe focus in general more on the interannual variability? You state that this could not really be studied before, so that is new. The above part of the introduction explains that the seasonal evolution seems to the strongly related to the seasonal variability in the wind field and this was known already before then?

Line 96-99: Maybe then also mention the relation here? They found the intensity to be related ... and the core position to be related ...

Line 105: I again miss that the STCs are mentioned as part of the circulation system e.g. see Tuchen et al. 2019: https://doi.org/10.1029/2019JC015396
Tuchen et al., 2020: https://doi.org/10.1029/2020JC016592

Fig. 1 How is this figure obtained? Is really only literature taken here as an underlying base for this? Or is Figure 2a the base for this? To my knowledge such a schematic should always be based on observed data, then also the length or width of the arrows can be somewhat adjusted to the observed current strength giving this a stronger basis then literature alone. Surely Fig. 2a does not include subsurface currents as EUC/NBUC so these are definitely just added because of literature, I guess.

Line 163: Maybe also state the rotation angles you used for the velocities here, if someone would want to compare results.

Fig. 2b-d: I am not quite sure whether there is a more sophisticated method to look at this, but it seems to give reasonable results. However, the figures are really small and it is hard to read the numbers. Is it maybe possible to use the same color scale?

Section 2.2. and Section 2.3: See my comment above; how well do these products fit together?

Line 229: at every grid point? And then compared the peaks? I am not quite sure what is done here?

Line 243: So interannual variability is only important in a very small area? Does that mean that investigating the interannual variability of the whole region is somewhat not necessary? Is that maybe an important result? But this then means that the new thing to look at interannual variations in the region is only important for a really small region?

Fig 3: For me it was hard at first to understand what is plotted here. I thought the ticks on the right of the contour plot would belong to the left hand contour plot as well and was quite irritated at first what is plotted here. For me ticks should never have 3 digits after the dot as it makes the numbers too huge. The colorbar is somehow in the plot. Green line is hardly visible. Explanation about the mean on the far left seems rater complicated in the caption. The left subplot is just the mean over the timeseries right? Why are there no data shown for section 5 in the southern region? I think the ylimits are wrong. They are the same as for section 6 although section 6 extends further south.

Line 284: the seasonal and **to** a lesser extent …

Line 289: … an eastward **surface** current, where …
State here where exactly the EUC is located in this region, from which longitude is it reported until where and what is its depth extent to clarify this surface current is distinct from the EUC.

Line 378: Does that mean that your definition of the current width includes both cores then in Table 1?

Fig: 4 Why is the map of sections shown again? It is already indicated in Fig .1. and so small that anyway nothing can be seen on the figure. I suggest to remove that subplot and to enlarge the other subplots as it is very hard to read the ticks in the figures. I would put in the caption directly that this is the average seasonal cycle obtained from Figure 3 as stated in the text Line 409-411. Then you can remove that there and just talk about your results.

Fig. 5 somehow it appears odd that the NBC changes width and strength in this way. Is this somehow possible to explain. Does that have to do with the definition and the fact that there are maybe some rings present at the NBC2 location?!
Maybe this will get discussed. 1 & 3 seem in phase and 2 and 4 although 2 and 4 are in the same eddy driven regime right, so maybe this should be discussed here.
Is there somehow a better way to show all these curves or maybe do not show them all, but concentrate on the ones necessary for the key points. I find it rather hard to understand all the different curves with all these acronyms being further extended by additional letters. Also, I mixed up the lines for the core velocity and the location of the core. Maybe make current wise subplots?

Line 477: Is this really a continuous flow along the coast? Isn't this the track also of the NBC rings? There might appear a continuous flow in the average, but in practice the transport is accomplished by these large vortices, which are not wind-driven, but forced as you state by barotropic and baroclinic instabilities. Hence it is clear, that this does not follow the wind?!

Line 481: Here you mention the rings. I think you have to discuss this more clearly.

In general, all these acronyms make you go crazy, but I do not have a good suggestion about how to change this.

Line 504: What do you mean here? Eastward surface flow corresponds to the mixing of subsurface flow? I do not really understand what you want to say here.

Line 519-520 nSEC mixes with NEUC? For my understanding the NEUC is a subsurface flow. Does it have a surface extension then at this time? Otherwise, I do not quite understand how nSEC and NEUC are supposed to mix. Please clarify.

Line 529-530: Isnt that obvious? For my understanding you only have the nSEC because the Trades traverse the equator as the ITCZ is positioned always north of the equator. The seasonal migration of the ITCZ therefore mainly affects the seasonal cycles for the nSEC and the NECC and not as much for the other parts of the SEC or not? I think you should always

discuss the seasonal cycles you obtain in conjunction with the changes in the large scale circulation.

Line 530: the nSEC does not cross the S4 section at a right angle. Couldn't that also have imprints on not being able to correctly determine the seasonal cycle of the intensity? Okay this is mentioned in line 542. I would maybe point this out beforehand and in any way, I think for all this description maybe think about what is important to say and what is maybe not; in this case maybe do not consider the SEC part from S4?

Section 4.4: See my general remark. You have to clarify how trustworthy your geostrophic surface velocities are in close proximity to the equator and whether then this current "X" is real or also could be an artefact.

Figure 6 is more what I would envision for Figurer 5 on seasonal scales concerning a subplot for each current vein.
Maybe better to remove the seasonal cycle in Figure 6 though to better see interannual variations? Its rather hard to see anything beyond the seasonal signal in these time series; this seems to have been done in Fig. 7 for some of the currents. Maybe somehow combine Figure 6 and 7?

Line 592ff: What do the deviations stand for? Standard deviation? Standard error? Why do you state all these numbers in the text? What do we learn from that?

Line 599: as mentioned above this should be clearly seen when the seasonal cycle is removed.

Line 600: Isnt it obvious that they should be correlated, if the NBCR feeds the NECC? In any way which time series did you correlate. I thought you want to look at the relation on interannual time scales, but I think that a 3 month running mean does not remove the seasonal component. Along that lines before you say that interannual variability is only important in small region which should maybe imply that the correlations to the other current bands on interannual time scales must be very small? I think this needs clarification.

Line 617-619: Didn't you just say that the NECC and the NBCR were correlated and now no obvious relation can be seen. Or maybe you mean no relation between core position and strength is obvious? I think you have to rephrase this to make clear what is meant.

Line 620: Would you expect such a relation necessarily? What would be the mechanism pushing the core to a certain position and in the same time influencing the intensity?

Line 624ff: Here I find more methods/analysis are necessary, see my general comment above. You cite Hormann et al. 2012, where an extensive analysis of the interannual NECC variability and its relation to the Atlantic climate modes was made. Along that line: What are your new results about this? In Hormann et al. 2012 they use CEOF analysis, regression and composite analysis to determine this relationship.
Here only correlations of different timeseries is used, which I think is not sufficient as this extensive study about interannual NECC variability and its relation to the climate modes (Hormann et al. 2012) already exists.

Line 625: I guess you mean 1993-2017?

Line 628: What to you learn about the spatial characteristics of the interannual variability in the study area when you only correlate time series? See my comment above I think you need some sort of regression and composite analysis for that.

Table 2: In the beginning of your study you show that interannual variability is basically only elevated in the NECC region. Please clarify for what kind of correlation you investigate the time series, this somehow does not get clear for me.

Line 701: I am not sure I understood everything. Maybe instead of all this neg/pos decrease/increase describe the relation in one way and then say vice versa?
But do I understand correctly that all these relationships between the currents and climate modes were already found and explained? So, what is new? Please highlight this more clearly.

Figure 8: I do not understand what is shown here in color. Probably you mean it is speed = sqrt (u^2+v^2) multiplied with the sign of the **zonal** velocity or not? If there is also the sign of meridional component in it would be confusing, but the plot looks like zonal?

Line 744-751: I think I do not understand this? So absolute velocities is something you do not have right. They should be something like Ekman+geostrophy, but again keep in mind that both a problematic at and close to the equator. Is this part then something like a validation that you compare this with other studies and say that they are maybe somewhat weak here? This could then also mean that your current X is not really trustworthy then? See my general remark about that.

Line 752-755: but only the geostrophic component of this flow and again its maybe not well defined in this product.

Fig. 10 so in this case the schematic is based on Fig. 8 as hopefully Figure 1 is based on 2a, but you need to keep in mind that you only show the geostrophic component! Or did you combine fIgure 8 and 9 to come up with Fig. 10? You can not really add the NBUC und EUC in this case as you have not included them in Fig. 8; you have the surface geostrophic currents, so the NBUC und EUC are not part of it. Why are there 3 arrows now for the NECC. I thought you divide into nNECC and sNECC what is the middle one for then?

Line 841-843: Then maybe some regression for the individual branches should be performed here to see whether the results of Hormann et al. 2012 hold for the individual parts or not?

Line 852: interannual variations only important in the Northeast I thought?

---

## Author Response (AR1)

**Response to Reviewer 1:**

Review of Dimoune et al.: "Revisiting the tropical Atlantic western boundary circulation from a 25-year time series of satellite altimetry data"

This study uses 25 years of satellite altimetry data to describe the mean surface circulation of the Western Tropical Atlantic. The authors describe in detail the seasonal cycle of different branches of the South Equatorial Current, the North Brazil Current and the Guyana Current at different zonal and meridional transects. A main novel result of this manuscript is the description of a current branch at 0-2∘N, above the Equatorial Undercurrent. This surface branch was previously unremarked in the literature and appears to be an extension of the North Brazil Current retroflection. Consistent with the literature, the results show that some of these branches have in-phase seasonal cycles, peaking in late winter/early spring, whereas others peak in fall. The interannual variability of the circulation is related to the Tropical Atlantic Meridional Mode. This is study is a relevant update of the mean and low-frequency variability of the surface circulation in a globally important region of the tropical ocean, where vigorous interhemispheric exchanges take place. The analysis is simple but robust and the manuscript is organized logically. I have to admit, however, that I had a hard time getting through the text, which is terse and acronym-laden. Besides addressing the technical points below, I strongly recommend the authors work on their text to make it accessible to those who do not work on the Tropical Atlantic oceanography every day.

Answer: First of all, we thank you for your review, comments and remarks that help to improve this work and make it clearest and the understandable possible.
We have carefully taken in consideration the comments and remarks. The complexity of the western boundary and the multiple currents that are involved, and which are studied here can make the text hard to be accessible. However, we have tried to make it more understandable by creating a table of acronyms to help the reader to rapidly refer to if necessary.

Specific and technical points:

1. Geostrophic currents near the Equator
The manuscript lacks a description of the robustness of the equatorial β−plane approximation for calculating geostrophic velocities at • }3∘ of the Equator. How accurate are these velocities? Can you really trust small changes in speed across the Equator (e.g., described in section 3)?

Answer: Thank you for your comments. Indeed, you are right. So, in the new version of the manuscript, we have given more details about the β−plane approximation. We have also looked for current meter and ADCP data to validate the geostrophic currents at the equator, and prove the existence of the equatorial eastward surface flow. We found at the PIRATA buoy location available in our study area (0°N35°W) current meter observations at 12-m depth, corresponding to a few months in our data time series (11/10/2017-29/01/2018: Figure 2 below) that we have used to validate the geostrophic currents in the equatorial region. We also looked for German cruises SADCP data (downloaded from the data center PANGEA https://doi.pangaea.de/10.1594/PANGAEA.937809 and described in Tuchen et al, 2022) and plotted the meridional sections of the currents at 40°W, 35°W and 32°W to show the existence of the equatorial surface eastward flow in the surface.
Here below are some analyses done:
The Comparison between the current components from the current meter at 12-m depth and the geostrophic currents (interpolated to the equator) over the period 11/10/2017-29/01/2018 (5 days means) shows as usual an underestimation of the latter (Picaut et al., 1989; Lagerloef et al., 1999, Pujol et al., 2016). The zonal components of the currents from both data show a correlation of 0.71 while the meridional ones are weaker (Figure 1a-b). The mean biases/standard deviation errors of both components are respectively 0.04 /0.11 m s$^{-1}$ and 0.14/0.03 m s$^{-1}$. This result is consistent with Lagerloef et al. (1999) who found similar value in the western Pacific (0°N165°E and 0°N170°W). The authors have compared current mooring (10-m depth zonal component) to the zonal component of the geostrophic current at the equator and found correlations of ~0.70 and biases <0.1. Our results compared to the previous ones give then credit to the altimeter-derived geostrophic currents used in this study.
Looking forward to investigate the surface eastward currents the available data of the PIRATA current meter from November 2018 to 25/03/2019 show a surface eastward current at 12-m depth during the whole period (Figure 2). This confirms the previous findings of Bourlès et al. (1999b) and justify our investigation to know more about this surface current.

[Figure]

Figure 1: Time series of the PIRATA mooring current (12-m depth) and the altimetry geostrophic currents at 0°N35°W over 11/10/2017-25/03/2019 period. The top and middle panels represent the zonal and the meridional components of the currents, respectively, and the bottom panel represents the current speed. Note that the geostrophic currents were only available for 11/10/2017-29/01/2018 period.

We have also analyzed all the individual Shipboard ADCP sections from German cruises available in the study area (meridional sections at 40°W, 35°W and 32°W) to look for the presence of the equatorial surface eastward flow shown in our study. The first depth at the surface of each section varies from 0-m to 17-m depth depending on each cruise, and we have now taken them into account in the new version of the paper to argue about the presence of the surface eastward flow in our study area. Figures 2-10 below show both the zonal and meridional components of the currents (top and bottom panels, respectively). The dashed/solid contours represent the westward/eastward currents, and the contour intervals are each 0.2 m/s for both components U (zonal) and V (meridional).

 The sections at 40°W during the first half of the year (Figures 2-3) clearly show the presence of an eastward flow in the upper layer between 0°-2°N as shown using the geostrophic currents. At 35°W, this flow is extended to 2°S, with usually a northward meridional component between 0°-2°N (Figures 4-7) in the first half of the year (March-June). This is consistent with the

cyclonic circulation found using the geostrophic current in our study during boreal spring. In October-November (Figures 8-9), the equatorial surface eastward flow appears weaker and less extended (1°S-1°N) with a southward meridional component between 0°-1°N. This may explain why we didn't find any cyclonic circulation during the second part of the year. At 32°W (Figure 10), the unique section in June show a weaker surface flow in the upper layer shifted to the south between 2°S-0°N. This is also consistent with our findings.

[Figure]

Figure 2: Shipboard ADCP section at 40°W between 2°S-2°N during 07/03/1994 to 10/03/1994 period.

[Figure]

Figure 3: Shipboard ADCP section at 40°W between 2°S-2°N during 03/05/2003 to 05/05/2003 period.

[Figure]

Figure 4: Shipboard ADCP section at 35°W between 6°S-3°N during 30/05/1991 to 05/06/1991 period.

[Figure]

Figure 5: Shipboard ADCP section at 35°W between 5°S-5°N during 13/03/1994 to 18/03/1994 period.

[Figure]

Figure 6: Shipboard ADCP section at 35°W between 6°S-8°N during 09/05/2002 to 16/05/2002 period.

[Figure]

Figure 7: Shipboard ADCP section at 35°W between 5°S-3°N during 26/05/2006 to 01/06/2006 period.

[Figure]

Figure 8: Shipboard ADCP section at 35°W between 6°S-0°N during 17/10/1990 to 22/10/1990 period.

[Figure]

Figure 9: Shipboard ADCP section at 35°W between 5°S-4°N during 02/11/1992 to 07/11/1992 period.

[Figure]

Figure 10: Shipboard ADCP section at 32°W between 2°S-2°N during 12/06/2006 to 13/06/2006 period.

Reference: Tuchen, F.P., P. Brandt, J.F. Lübbecke, et R. Hummels, Transports and Pathways of the Tropical AMOC Return Flow From Argo Data and Shipboard Velocity Measurements,

Journal of Geophysical Research: Oceans, 127 (2), e2021JC018115, https://doi.org/10.1029/2021JC018115, 2022

2. Intraseasonal variability

From figure 2, the intraseasonal/subsesonal variability (< 120 days) is as important as the seasonal variability. The authors should discuss the intraseasonal variability instead of the interannual variability.

Answer: Thank you for your suggestion. We have chosen to dedicated our study to the seasonal and interannual variability because recent studies have been already done concerning the intraseasonal variability with also products derived from altimetry and multi-observations products. However, as we mentioned at the end of the paper, our new challenge would be to find a coastal altimetry product that will allow a significantly better resolution and accuracy along the continental shelf to further explore this question.

3. Some methodological details are missing

The authors should improve their description of their estimation of various current properties. For example, how do the authors estimate these properties in table 1? Are the authors simply choosing are maximum velocity in the transect? How about the current width? Are the authors eye-balling this property from year transect? Wouldn't fitting a functions (e.g., a gaussian) to the cross-track velocity profile be a more effective way to estimate those properties? Also, does the interpolation of satellite altimetry sea-level from along-track to regular grid smear the currents? What's the effect of this interpolation on the width and intensity of the currents? Please, discuss.

Answer: Thank you for your suggestion. We have tried to give further description in the corrected manuscript in order to facilitate the reading.
Indeed, our approach to determine the current width was the simple one. The width of the currents is determined by the mean of the current over the whole period of study (1993-2017) at each grid point. Then, it is important to know the sign or direction of the current (eastward or westward) to be able to define the limits of the currents which are the location where the current is null. So, the distance between both limits is define as the width of the currents (See left boxes of each Hovmöller diagram in Fig. 3). However, to compute the strength of the currents, our approach was to consider the flow between the contour lines of half of the

maximum speed (velocity) from either side, and average the values over time to obtain time series.

It has been reformulated in the revised manuscript as follows: "In order to further investigate the temporal variations of the current amplitude or strength, the maximum speed (velocity) values of each current paths have been determined for each month. These maxima correspond to the current core and are called Vmax hereafter (see Fig. 3, green line on S1), and their location corresponds to the location of the current core. Then, to estimate the current relative strength/intensity, we have considered part of the sections of velocities larger than Vmax/2 (see Fig. 3, blue lines on S1); and we finally computed the strength/intensity values by averaging all the velocity values of these parts over time. The width of the currents (Table 2) is determined by the mean of the current over the whole period of study (1993-2017) at each grid point (see left boxes of the Hovmöller diagrams of Fig. 3). Then, knowing the sign or direction of the current (eastward or westward), the corresponding time series are used to define the width by finding the limits where the current is null. Note that the current strength/intensity has also been computed by averaging velocity values over the entire width of the current paths (e.g., not only where the velocity is larger than Vmax/2), but the results did not correctly reflect the current variability observed in Fig. 3 and Fig. 4 (not shown).

For the NBCR and the NECC, the variability of the maximum speed (velocity) of apparently two flow/branches and their corresponding locations is also analyzed, in order to compare with the study of Fonseca et al. (2004). The goal is also to investigate the variability of the current location with respect to the wind variability and to the tropical Atlantic climate modes. In this study, the presence of apparently two flows of the NBCR and the NECC two-core structure are identified when the velocity profile of the currents shows two local maxima separated by a local minimum respectively, in the NBCR region (considered between 4°-10°N) and the NECC region (considered between 3°-11°N) (Fig. 4S4-S6)."

Concerning the interpolation, so far, the method used to produce the DUACS (Data Unification and Altimeter Combination System) products is the best compare to the other and is known to minimize the error effect on the derived data (Lagerloef et al., 1999; Dibarboure et al., 2011; Pujol et al., 2016).

New reference: Dibarboure, G., Boy, F., Desjonqueres, J. D., Labroue, S., Lasne, Y., Picot, N., Poisson, J. C., and Thibaut, P.: Investigat- ing Short-Wavelength Correlated Errors on Low-

Resolution Mode Altimetry, J. Atmos. Ocean. Technol., 31, 1337–1362, doi:10.1175/JTECH-D-13-00081.1, 2014.

4. Figures are difficult to read

Most figures are barely readable. The labels are tiny and oftentimes there are too many lines in the plots. The authors should increase the labels, rearrange the panels, and try to use less lines to improve readability.

Answer: Thank you for the suggestion. We have improved the figures. For the case of Fig. 4, we even removed the map at the left side, because the reader can refer to Fig. 1.
Here below is the new Fig. 4:

[Figure]

**Figure 4.** Average seasonal cycle obtained from Figure 3 (vectors of the currents are superimposed on the contour of their amplitude in m s$^{-1}$). On the right sides of each subplots, the distances from the southernmost point (in km) are indicated.

5. Tidal correction on the Amazon Shelf

I think the "erroneous altimetry measurements" mentioned around line 160 is actually poor tidal corrections on the Amazon Shelf. The authors should more clearly described what they consider unrealistic values. What criterion are the authors using to remove those values?

Answer: Thank you for the comment. Yes, we have made it clear in the corrected version of the manuscript. We have reformulated as follows: "Note that, data with higher variability (standard deviation > 0.4) have been removed. They are found in the Amazon region which is not a

primary area of interest for this study, and where, the annual mean current speeds are unrealistic (higher than 2.5 m s$^{-1}$), probably due to geographically correlated errors (Pujol et al., 2016).

6. Wind-driven currents

As the name suggests, the GEKCO product contains both geostrophic and Ekman currents. If the authors are interested only in Ekman currents for figure 9, why don't the authors calculate those directly from ERA5?

Answer: Yes, you are right. In fact, the GEKCO Ekman currents were only used to evaluate the probable importance of the Ekman currents over the altimeter-derived geostrophic currents in the equatorial region. We chose to use them because they have been already validated in the equatorial region (Sudre et al., 2013), and we mentioned it to inform and notify the work that has been done.

As the goal of the paper was to evaluate the influence of the large-scale remote wind on the regional circulation, the use of ERA5 wind is justified. Also, these currents do not impact much the geostrophic currents west of 32°W. So, we decide to remove it in the final version of the manuscript to avoid confusions since the geostrophic currents are validated in the equatorial region.

Typing, English and minor technical corrections
    1.  line 43: change works with studies.

Answer: Thank you for your comment. It has been considered.

    2.  line 51: (...) North Brazil Under Currents (NBUC) which raises to the surface around → (...) North Brazil Under Currents (NBUC), which surfaces around.

Answer: Thank you for your comment. It has been considered.

    3.  line 77: meaning of "no more respected" is unclear. Do you mean "no longer satisfied"?

Answer: Thank you for your comment. Indeed, we wanted to say "no longer satisfied". The sentence has been reformulated.

4. line 92: Do you mean understudied?

Answer: Thank you for your comment. Indeed, we wanted to say understudied. It has been considered.

5. lines 108, 110 and elsewhere: "to allow" is a transitive verb—something or somebody allows somebody to do something else. So the dataset allows you to provide a more robust (...).

Answer: Thank you for so much for your comment. It has been considered.

6. line 111: as follow → as follows.

Answer: Thank you for your comment. It has been considered.

7. line 125: Why the hat in Goˆes?

Answer: Thank you for your comment. It was a mistake. It has been considered.

8. line 158: averaged on a monthly basis → averaged monthly.
Answer: Thank you for your comment. It was a mistake. It has been considered.

9. line 183 and elsewhere: Pound → Pond. (Also, those should be 10-m winds, right?)

Answer: Thank you for your comment. Of course, it is 10-m winds. It has been considered.

10. line 184/eq. (1) and (2): plese use $\tau$, not $\zeta$, to refer to wind stress. Also, x and y in eq. (3) should in subscript: $\tau x$ and $\tau y$.

Answer: Thank you for your comment. It has been considered.

11. lines 196-197: why use zonally average wind instead of local winds?

Answer: We have average zonally because here, the goal was to assess the influence of the large-scale remote wind forcing on the regional circulation.

12. line 218: why is Guyana in figure 2?

Answer: Thank you for your comment. We have changed it, and said "farther north".

13. line 221 and elsewhere: Do you mean path instead of vein?

Answer: Yes, it was "path". We have changed it.

14. line 261: more than → longer than.

Answer: Thank you for your comment. It has been considered.

15. line 261: less than → shorter than.

Answer: Thank you for your comment. It has been considered.

16. Figure 2: you should mention in the caption that the colorbar of (c) is different than the ones of (b) and (d).

Answer: Thank you for your comment. It has been mentioned now.

17. line 291 and elsewhere: to name a current or current branch X is a horrible idea. Please, be creative and come up with a more descriptive name.

Answer: Thank you for your encouraging comment. We named it the Equatorial Surface Eastward Flow. However, because the multiple acronyms in the text, to simplify things, we called it X.

18. line 393: lowest → southernmost.

Answer: Thank you for your comment. We have changed it.

19. line 413: remove extra parenthesis.

Answer: Thank you for your comment. It has been removed.

20. line 428: currents intensity → speed.

Answer: Thank you for your comment. We have changed it.

21. line 484: Further north → Farther north.

Answer: Thank you for your comment. It has been considered.

22. line 517: recirculate → recirculates.

Answer: Thank you for your comment. It has been modified.

23. line 591-592 and elsewhere: what does • } stands for here? Standard derivation? Standard error?

Answer: It stands for the standard deviation.

23. line 630: Student test → Student's test.

Answer: Thank you for your comment. It has been modified.

24. line 670: analysed → analyzed.

Answer: Thank you for your comment. It has been modified.

25. line 743: ADCPs measurements → ADCP measurements.

Answer: Thank you for your comment. It has been modified.

26. line 753: similar with → similar to.

Answer: Thank you for your comment. It has been modified.

27. line 870: he → the.

Answer: Thank you for your comment. It has been modified.

28. line 880: confirm → confirms.

Answer: Thank you for your comment. It has been modified.

**Response to Reviewer 2:**

**Review of "Revisiting the tropical Atlantic western boundary circulation from a 25-year time series of satellite altimetry data" by Djoirka M. Dimoune, Florence Birol, Fabrice Hernandez, Fabien Léger, Moacyr Araujo [Paper # egusphere-2022-402]**

**The study uses a gridded product of geostrophic surface velocities based on satellite altimeter observations to determine the seasonal and interannual variability of the surface circulation in the western tropical Atlantic. In addition to this surface velocity product, Ekman currents are used from another gridded product (GEKCO) as well as gridded surface winds from ERA5 in order to relate the geostrophic surface velocity variability to changes in the wind field. Furthermore, a sea surface temperature product is used to investigate the relation of the surface velocity variability to variability associated with the Atlantic climate modes, the meridional as well as the zonal mode.**

**In order to investigate the seasonal and interannual variability of the individual current branches, 6 sections are defined in order to determine the strength and position of the individual current branches and to compare their variability patterns among each other. The results show that a large fraction of the currents show a similar seasonal cycle, while a second regime seems to exist showing a somewhat different seasonal evolution, but agrees amongst each other.**

**An eastward current within the equatorial region and west of 42°W is identified and related to a larger scale cyclonic circulation.**

**The NECC is found to have a double core structure, which is somehow related to seasonal chances in the wind stress curl.**

**Interannual variations of the NBC seem to be opposite in the different hemispheres, which are related to the Atlantic meridional mode phases as well as the interannual variability of the NECC and nSEC at 42°W. Interannual variability of the NECC and nSEC at 32°W is related to the zonal mode phases.**

**Formal review:**

**The manuscript is well written and organized. Understanding the seasonal to interannual variability of the western tropical Atlantic surface circulation and its forcing mechanism is essential in order to understand the transport of heat, salt and other tracers across the equator and hence between the hemispheres and therefore definitely deserves attention. The methods used to investigate the seasonal to interannual variability are well explained,**

however I miss a more sophisticated method to corroborate the relation of the assessed variability to the forcing mechanisms in terms of the variability in the wind field and the different phases of the Atlantic climate modes. This study cites Hormann et al. 2012, where a lot of effort is put into the investigation of this relation e.g. regression and composite analysis etc. Maybe something like this could be included here. In my opinion as gridded products are used here only and hence no need for own calibration or gridding methods, more effort should be put in the validation of these products with other available data sets.

In total I find the subject of the study important, but I recommend that some additional methods should be included in order to corroborate the findings of the study.

**Answer:** First of all, we thank you for your review, comments and remarks that help to improve this work and make it clearest and the understandable possible.

The goal of the paper was to take advantage of the longer time series of observations derived from altimetry data (1993-2017) to revisit the tropical Atlantic western boundary circulation by doing simple but significant analyses to study its spatial and temporal variability. In this paper, we dedicated our study only the seasonal and the interannual timescale because some recent works have been done already with the altimetry or multi-observations data at the intraseasonal timescale (eg., Aguedjou et al., 2019 and Aroucha et al., 2020).

The study was motivated by the following remaining questions in the study area:

➢ How does the North Brazil Current (NBC) behave along its pathway from south to north of the equator?

➢ What are the probable relationships between the NBC and the other surface currents, and between all these currents, the large-scale wind variability and the tropical Atlantic modes on respectively, the seasonal and interannual timescales?

By this study we wanted also to investigate with the new altimeter-derived product, the probable presence of the equatorial surface eastward current mentioned in the literature by Hisard and Hénin (1987) and Bourlès et al. (1999b).

To do so, our first objective was to use a new approach of calculation of the current's strength/intensity (based on the mean of current speeds higher than the half of the maximum speed in the velocity field) to study the temporal variability of the strength of all the currents involved in the western boundary circulation in comparison to their volume transport mentioned in the literature. To follow their spatial variability, 6 sections crossing the currents of interest

are used to track the possible changes in the currents along their pathway, which is difficult to be highlighted in the overall previous studies. Among the sections, we chose one section crossing the North Brazil Current Retroflection area to also track the spatial variability of the retroflection location and compare with previous estimations.

The second objective was to investigate the relationship between the currents' variability, the large-scale trade wind field variability in the eastern basin, and the tropical Atlantic Climate modes (zonal and meridional).

As you mentioned, the biggest challenge of the geostrophic currents derived from altimetry is the question of their reliability in the equatorial region. The challenge is even bigger in the Atlantic Ocean where there are less observations (in situ, drifters: See Figure 1 below). So, we built our study on what was already done in the Pacific, considering the fact that most of the variance of the ocean currents could be explained by the geostrophic and the Ekman currents. Ralph and Nüler (1999) have shown that almost 80% of the variance of the currents observed by Lagrangian drogue drifters at 15-m depth is explained by both components: 63% for geostrophic currents and 15% for Ekman man currents. Especially for the zonal currents, Lagerloef et al. (1999) found that, in the tropical Pacific, the mean zonal currents were dominated by the geostrophic term.

For a reliable study, it was very important to choose the right product which should be closer to the reality. The choice of SEALEVEL_GLO_PHY_L4_REP_OBSERVATIONS_008_047 product is guided by the fact that in the equatorial region the currents have been calculated using the Lagerloef methodology (Lagerloef et al., 1999) which used a β-plane approximation where the Coriolis parameter is vanishing. The method was improved in this product compared to the previous ones by introducing the meridional velocities in the β component leading to more intense currents and improving the continuity of the currents within the latitudes $\pm 5\circ$ N (See Pujol et al., 2016). The gridding procedure is so far the best used compared to the others (eg., Optimum Interpolation), and the performance of the grid is proved to reproduce reasonably the velocity fields (Dibarboure et al., 2011; Pujol et al., 2016).

The GEKCO Ekman currents were only used to evaluate the probable importance of the Ekman currents over the altimeter-derived currents in the equatorial region. We chose to use them because they have been already validated in the equatorial region (Sudre et al., 2013), and we mentioned it to inform and notify the work that has been done.

To investigate the relationship with the winds, we have used the ERA5 wind velocity components to compute the wind parameters (The location of the Intertropical Convergence Zone: ITCZ, the wind stress curl Strength) according to Fonseca et al. (2004). The Atlantic climate mode indices have been computed according to Servain (1991) and Zebiak (1993) to understand the possible relationship with the interannual variability of the currents.

[Figure]

Figure 1: Status of Global drifter array on November 21, 2022, taken from NOAA website (https://www.aoml.noaa.gov/phod/gdp/)

**General remarks:**

As mentioned before only gridded products are used here, which are readily available. However, these products based on geostrophy and Ekman all need to be adapted within the equatorial region as geostrophy and Ekman do not hold there due to the vanishing Coriolis parameter. I think more effort and/or explanation have to be included to convince the reader that these products are trustworthy in this region e.g. maybe compare the equatorial velocities to surface velocities measured at the PIRATA buoy sites? Along these lines which data set can be used for comparison of the Amazon outflow? It is just stated

that the values of the surface velocity product there are unrealistic, but it needs some sort of reasoning to say so, before just blanking out these values.

Another point about the products: An Ekman velocity product is used, which is available and then the wind stress is calculated from the ERA5 winds. However, calculating the wind stress relies on empirical functions as explained in Line 184 and hence is crucially dependent on the formula used. In my opinion this means using an Ekman product on the one hand and calculating wind stress from some wind product on the other hand could lead to wind stress and Ekman product not necessarily fitting together, which is maybe not so nice when interpreting the results or at least this should be clearly stated somewhere. Are e.g. the empirical functions for wind stress to determine the Ekman products the same or do they differ?

To me it gets not exactly clear what are the new results from this study. For some of the results it is stated that it agrees with former studies and I might also miss some literature information, but you should definitely make point out more clearly what are the new findings here.

The detection of current X seems a somewhat new result as I understand. However, it is based on velocities from the equatorial band, which are critical for the products in use in this region (see my comment above). Hence, I think it is crucial at this point to validate the products in the equatorial region in order to make clear that this current X is not an artefact of the method to obtain the velocities in this region.

Answer: We agree with the general remark. Due to the lack of observations in the equatorial region (Figure 1 above), we initially chose to rely on the altimetry geostrophic currents. But as you suggested, we were able to find one PIRATA mooring in the equatorial region (0°N35°W) in our study area where some currents data were available only for a few months in our data time series (11/10/2017-29/01/2018: Figure 2 below).

The Comparison between the current components from the current meter at 12-m depth and the geostrophic currents (interpolated to the equator) over the period 11/10/2017-29/01/2018 (5 days means) shows as usual an underestimation of the latter (Picaut et al., 1989; Lagerloef et al., 1999, Pujol et al., 2016). The zonal components of the currents from both data show a correlation of 0.71 while the meridional ones are weaker (Figure 2a-b). The mean biases/standard deviation errors of both components are respectively 0.04 /0.11 m s$^{-1}$ and 0.14/0.03 m s$^{-1}$. This result is consistent with Lagerloef et al. (1999) who found similar value in the western Pacific (0°N165°E and 0°N170°W). The authors have compared current mooring

(10-m depth zonal component) to the zonal component of the geostrophic current at the equator and found correlations of ~0.70 and biases <0.1. Our results compared to the previous ones give then credit to the altimeter-derived geostrophic currents used in this study.

Looking forward to investigate the surface eastward currents the available data of the PIRATA current meter from November 2018 to 25/03/2019 show a surface eastward current at 12-m depth during the whole period (Figure 2). This confirms the previous findings of Bourlès et al. (1999b) and justify our investigation to know more about this surface current.

[Figure]

Figure 2: Time series of the PIRATA mooring current (12-m depth) and the altimetry geostrophic currents at 0°N35°W over 11/10/2017-25/03/2019 period. The top and middle panels represent the zonal and the meridional components of the currents, respectively, and the bottom panel represents the current speed. Note that the geostrophic currents are only available for 11/10/2017-29/01/2018 period.

We have also analyzed all the individual Shipboard ADCP sections from German cruises available in the study area (meridional sections at 40°W, 35°W and 32°W obtained from the data center PANGEA https://doi.pangaea.de/10.1594/PANGAEA.937809 and described in Tuchen et al, 2022) to look for the presence of the equatorial surface eastward flow shown in our study. The first depth at the surface of each section varies from 0-m to 17-m depth depending on each cruise, and we have now taken them into account in the new version of the

paper to argue about the presence of the surface eastward flow in our study area. Figures 3-11 below show both the zonal and meridional components of the currents (top and bottom panels, respectively). The dashed/solid contours represent the westward/eastward currents, and the contour intervals are each 0.2 m/s for both components U (zonal) and V (meridional).

The sections at 40°W during the first half of the year (Figures 3-4) clearly show the presence of an eastward flow in the upper layer between 0°-2°N as shown using the geostrophic currents. At 35°W, this flow is extended to 2°S, with usually a northward meridional component between 0°-2°N (Figures 5-8) in the first half of the year (March-June). This is consistent with the cyclonic circulation found using the geostrophic current in our study during boreal spring. In October-November (Figures 9-10), the equatorial surface eastward flow appears weaker and less extended (1°S-1°N) with a southward meridional component between 0°-1°N. This may explain why we didn't find any cyclonic circulation during the second part of the year. At 32°W, the unique section in June show a weaker surface flow in the upper layer shifted to the south between 2°S-0°N. This is also consistent with our findings.

[Figure]

Figure 3: Shipboard ADCP section at 40°W between 2°S-2°N during 07/03/1994 to 10/03/1994 period.

[Figure]

Figure 4: Shipboard ADCP section at 40°W between 2°S-2°N during 03/05/2003 to 05/05/2003 period.

[Figure]

Figure 5: Shipboard ADCP section at 35°W between 6°S-3°N during 30/05/1991 to 05/06/1991 period.

[Figure]

Figure 6: Shipboard ADCP section at 35°W between 5°S-5°N during 13/03/1994 to 18/03/1994 period.

[Figure]

Figure 7: Shipboard ADCP section at 35°W between 6°S-8°N during 09/05/2002 to 16/05/2002 period.

[Figure]

Figure 8: Shipboard ADCP section at 35°W between 5°S-3°N during 26/05/2006 to 01/06/2006 period.

[Figure]

Figure 9: Shipboard ADCP section at 35°W between 6°S-0°N during 17/10/1990 to 22/10/1990 period.

[Figure]

Figure 10: Shipboard ADCP section at 35°W between 5°S-4°N during 02/11/1992 to 07/11/1992 period.

[Figure]

Figure 11: Shipboard ADCP section at 32°W between 2°S-2°N during 12/06/2006 to 13/06/2006 period.

Regarding the Amazon region where the data were blanked, we have removed data with highest standard deviation (>0.4), considering that, in this region the North Equatorial Countercurrent

(NECC) and the NBC have the highest variability. The map of the annual mean current speed in the region blanked were also found unrealistic with velocity higher than 2.5 m/s. We could not find any data set to confirm this consideration, but as far as our main interest is the main currents it doesn't impact the rest of our analysis.

Concerning the use of the GEKCO Ekman velocity, as mentioned above, we have used it because they have been already validated in this region (See Sudre et al., 2013). We have used the ERA5 wind to investigate the relationship with the large-scale wind variability. And as mentioned in the section 2.3. of the paper, we have considered the region covering 6° S -16° N, and 30° W-0° W, following Fonseca et al. (2004).

Concerning what are the new results, we have now highlighted them starting from the abstract. The reason why it was stated sometimes that the findings agree with former studies is first because all the studies about the seasonal variability of the currents in this region have been done with time series less than 10 years, especially for the NBC transport (Fonseca et al., 2004; Garzoli et al., 2004). So, as mentioned in the beginning of the answers, by revisiting the western boundary circulation we are trying to check the aspects of the circulation by using a new approach and comparing with what have been already done. The new results here are:

- The different seasonal variability of the NBC found north and south of the equator considering the current strength/intensity as computed.
- The relationship between the currents and the large-scale wind variability at the seasonal time scale. In the same time, we have showed the variability of the location of the NBC retroflection (NBCR) and NECC branches maximum speed (core) can be impacted by the wind variability in the whole basin differently. Regarding the NBCR flow, it has been found a secondary flow from September to December when its main core is at its northernmost location.
- And the relationship between the interannual variability of the currents and the tropical Atlantic mode.

Because of the complex ocean circulation in the tropical western boundary, we have many acronyms that are difficult to deal with, especially for a reader who is not used to this study area. To facilitate the reading, we have made a table of acronyms which is easy to refer to during the reading.

Relatively to the sophisticated analyses to be applied, the goal of our work was to use sections to try to understand the variabilities of the current strengths in the study area with simple but

robust analyses. So, as it was mentioned in the paper, spectral analyses were applied on the study area to find time scales that need to be exploited. After the calculation of the current strengths/intensities and identifying some currents core location, we have obtained time series, and the study of the seasonal and the interannual variability has been based on them. Then, the correlations and the significative test were helpful to analyze the possible relationship with the wind and the tropical Atlantic modes unlike the study of Hormann et al. (2012) who focus only on the NECC region and applied the complex EOF analyses and related regression analyses to investigate the NECC's interannual variability.

**In addition to these general remarks, I have some detailed remarks throughout the text:**
**Detailed remarks:**

**Line 39-41:** What do you mean here? Return branch of the AMOC influenced by the wind? Maybe use the statement of Schott et al. 2005 here? "The surface circulation in this region is a superposition of the AMOC, the flow related to the Subtropical Cells and Sverdrup dynamics." In general, I miss that the STCs are mentioned here. They are shallow overturning cells connecting the tropics and subtropics and also have their imprint on the surface circulation.

Answer: Thank you for your suggestion. We have reformulated as follows: "It corresponds to a superposition of the return branch of the thermohaline Atlantic Meridional Overturning Circulation (AMOC), the flow from the Subtropical Cells and Sverdrup dynamics (Schmitz and McCartney, 1993; Schott et al., 2004, 2005; Rodrigues et al., 2007; Tuchen et al., 2019, 2020)."

Line 67-70 What about the formation of the NBC rings? This is also quite characteristic of the surface circulation and should be mentioned here.

Answer: Thank you for your suggestion. Here are the sentences we added: "The region is also known to be influenced by large mesoscale activities due to the barotropic instabilities of the currents. The most dominant mesoscale structures are large rings generated by the North Brazilian Current (NBC) retroflection (Aguedjou et al., 2019; Aroucha et al., 2020)."

Line 72: Which means in general Sverdrup dynamics apply here (why do you separate into Ekman and geostrophic velocities here?) superimposed by the AMOC and STC (see my comment above), even if west of 32°W this does not apply anymore as you mention later.

Answer: Yes, we agree. You are right. We have taken into account your suggestion and reformulated as suggested: "It corresponds to a superposition of the return branch of the thermohaline Atlantic Meridional Overturning Circulation (AMOC), the flow from the Subtropical Cells and Sverdrup dynamics (Schmitz and McCartney, 1993; Schott et al., 2004, 2005; Rodrigues et al., 2007; Tuchen et al., 2019, 2020)."

Line 86: response **to** the the wind stress curl

Answer: You are right. Thank you for the correction. It has been considered.

Line 92: So then should your study maybe focus in general more on the interannual variability? You state that this could not really be studied before, so that is new. The above part of the introduction explains that the seasonal evolution seems to the strongly related to the seasonal variability in the wind field and this was known already before then?

Answer: As mentioned in the general answer above, we wanted to revisit the whole western boundary circulation by using a new approach. So, the first thing to do was to take advantage of the longer time series available so far to understand the seasonal variability before investigating the interannual variability. As mentioned in the introduction, we know that the surface circulation basically influenced by winds. However, we wanted to understand how the large-scale wind variability impact the strength of the currents along their pathway (what was not yet done in the tropical western boundary).

Line 96-99: Maybe then also mention the relation here? They found the intensity to be related … and the core position to be related …

Answer: Thank you for your suggestion. Indeed, they found the northward shift of the NECC's core was associated with the warm phase of the Atlantic Meridional Mode (AMM) phased in boreal spring, whereas the NECC intensity was associated with the cold phase of the Atlantic Zonal Mode (AZM), phased in boreal summer. We have reformulated as follows: "They found

the intensity to be related to the cold phase of AZM and the core location to be related to the warm phase of AMM."

Line 105: I again miss that the STCs are mentioned as part of the circulation system e.g. see
Tuchen et al. 2019: https://doi.org/10.1029/2019JC015396
Tuchen et al., 2020: https://doi.org/10.1029/2020JC016592

Answer: Thank you for your suggestion. We have yet mentioned it in the first paragraph of the introduction and both papers have been cited. Indeed, the large-scale ocean circulation (subtropical cells) consists of the poleward Ekman transport at the surface, subduction in the equatorial in the subtropics, equatorward flow in the subsurface and upwelling along the equator and at the eastern boundary. As we are only focusing on the surface circulation, we did not want to last too long on it.

Fig. 1 How is this figure obtained? Is really only literature taken here as an underlying base for this? Or is Figure 2a the base for this? To my knowledge such a schematic should always be based on observed data, then also the length or width of the arrows can be somewhat adjusted to the observed current strength giving this a stronger basis then literature alone. Surely Fig. 2a does not include subsurface currents as EUC/NBUC so these are definitely just added because of literature, I guess.

Answer: Thank you for your remark. The figure is based on the literature. We have now reformulated better the figure title. As mentioned in the introduction, this figure aims to only show the currents involved in the western boundary circulations to introduce the region to reader before to start our study focusing on only the surface currents. Then, the strength of the currents will be discussed later, and a new seasonal map can be proposed by adjusting the width of the arrows to their observed strength.

Line 163: Maybe also state the rotation angles you used for the velocities here, if someone would want to compare results.

Answer: Thank you for your suggestion. It has been considered and the sentence have been reformulated as follows: "In this study, we considered only the cross-section component, and the rotation angles considered for the oblique sections S1-S2-S3/S4 are 45°/315°."

Fig. 2b-d: I am not quite sure whether there is a more sophisticated method to look at this, but it seems to give reasonable results. However, the figures are really small and it is hard to read the numbers. Is it maybe possible to use the same color scale?

Answer: Thank you for the remark. The figure has been improved. We did not choose another colour scale for the signal with periods larger than 600 days because we wanted to highlight clearly the interannual signal which appears to be weaker than the others as described in the paper.

Section 2.2. and Section 2.3: See my comment above; how well do these products fit together?

Answer: As mentioned in the first responses, the GEKCO Ekman currents have been chosen only because they have been already validated (Sudre et al., 2013), and we have used it only to evaluate the probable importance of the Ekman currents over the currents from the new altimeter-derived product in the equatorial region. After the validation work that has been done now, we did not find relevant to mention it anymore. So, we have removed the section 2.2 and the section 2.3 becomes the section 2.2. Consequently, the Figure 9 have also been removed.

Line 229: at every grid point? And then compared the peaks? I am not quite sure what is done here?

Answer: Thank you for your question. We have considered here the whole study area in which the currents have averaged in order have an overall view of the currents. Then, we computed the power spectral density of the corresponding daily averaged time series to find the dominant components of the currents in the study area.

Line 243: So interannual variability is only important in a very small area? Does that mean that investigating the interannual variability of the whole region is somewhat not necessary?
Is that maybe an important result? But this then means that the new thing to look at interannual variations in the region is only important for a really small region?

Answer: Thank you for your remark. You are right, the way we formulated the sentence was ambiguous. It is clear that, compared to the seasonal and the interannual signals, the interannual

signal is weaker. However, there are still things to be explored with the 25 years of data, even if the signal is strongest in the NECC area (See Fig. 3). The sentence has been reformulated as follows: "Intraseasonal fluctuations are also important in the same areas (with largest ratio of 0.44) while the interannual variability is weaker, and the highest values (>0.2) are found in the NECC area north/east of 4° N/40° W (consistent with Richardson and Walsh, 1986)."

Fig 3: For me it was hard at first to understand what is plotted here. I thought the ticks on the right of the contour plot would belong to the left hand contour plot as well and was quite irritated at first what is plotted here. For me ticks should never have 3 digits after the dot as it makes the numbers too huge. The colorbar is somehow in the plot. Green line is hardly visible. Explanation about the mean on the far left seems rather complicated in the caption. The left subplot is just the mean over the timeseries right? Why are there no data shown for section 5 in the southern region? I think the ylimits are wrong. They are the same as for section 6 although section 6 extends further south.

Answer: Thank you for your suggestions to improve the figure. It has been taken into consideration. The left subplots are the means over the times series. The section 5 was extended further south as for the section 6 to facilitate the comparison of the North Equatorial Countercurrent (NECC) along its pathway. As we can see in Fig 1., the section 5 is limited at 0 at 42°W. That is why there is no data.
The new figure is as follows:

[Figure]

Line 284: the seasonal and **to** a lesser extent …

Answer: You are right. Thank you for the correction. It has been considered.

Line 289: … an eastward **surface** current, where …
State here where exactly the EUC is located in this region, from which longitude is it reported until where and what is its depth extent to clarify this surface current is distinct from the EUC.

Answer: Thank you for the suggestion. It has been considered and the sentence has been reformulated as follows: "Note that in addition to the main currents mentioned in Fig. 1, between 2° S-2° N, the sections 4 to 6 show an eastward surface current, located at the same location where the EUC flows, west of 44°W in the subsurface along the equator (Schott et al., 1998). Depending on the season the EUC core is known to be located between about 100-m depth and 50-m depth when it surfaces to reach the near-surface (Brandt et al., 2016)".

Line 378: Does that mean that your definition of the current width includes both cores then in Table 1?

Answer: Yes, obviously.

Fig: 4 Why is the map of sections shown again? It is already indicated in Fig .1. and so small that anyway nothing can be seen on the figure. I suggest to remove that subplot and to enlarge the other subplots as it is very hard to read the ticks in the figures. I would put in the caption directly that this is the average seasonal cycle obtained from Figure 3 as stated in the text

Answer: Thank you for your suggestion. It has been considered and the new figure is here below:

[Figure]

**Figure 4.** Average seasonal cycle obtained from Figure 3 (vectors of the currents are superimposed on the contour of their amplitude in m s$^{-1}$). On the right sides of each subplots, the distances from the southernmost point (in km) are indicated.

Line 409-411. Then you can remove that there and just talk about your results.

Answer: Thank you for your suggestion. They have been removed.

Fig. 5 somehow it appears odd that the NBC changes width and strength in this way. Is this somehow possible to explain. Does that have to do with the definition and the fact that there are maybe some rings present at the NBC2 location?!

Maybe this will get discussed. 1 & 3 seem in phase and 2 and 4 although 2 and 4 are in the same eddy driven regime right, so maybe this should be discussed here.

Is there somehow a better way to show all these curves or maybe do not show them all, but concentrate on the ones necessary for the key points. I find it rather hard to understand all the different curves with all these acronyms being further extended by additional letters. Also, I mixed up the lines for the core velocity and the location of the core. Maybe make current wise subplots?

Answer: Thank you for your relevant questions and comment. We have considered them and tried to respond to them in the paper as follows: "Apparently, the NBC2 width and strength seems to be linked to the nSEC intensity in the eastern basin (see nSEC6 and nSEC4 in Fig. 5b). When the nSEC is weaker, the NBC2 is also weaker and narrower. Then, the NBC2 starts growing one month later when the nSEC intensity is increasing. This shows the importance of

the nSEC contribution to the NBC to the north of the equator. The delay between the nSEC and NBC2 growth can be due to the mesoscale activities more intense in the western basin (Aguedjou et al., 2019). The fact that the NBC1 and the NBC3 are in phase when the nSEC contribution to the NBC2 is lower might indicate that the NBC is more stable when the intrusion of water from the nSEC is weak. And then, most of the NBC in the upper layer tends to follow its northward route. Contrariwise, when the current is unstable because of the barotropic and baroclinic instabilities generated by the nSEC, most of the NBC flow tends to retroflects eastward. This can justify the phase of the NBC2, NBC4 and rNBC2."

Concerning the figure, it was difficult to find the good way to do it without losing the main information that we are trying to deliver. However, the idea of the acronyms table introduce in the new manuscript can be useful for the reader to understand easily.

Line 477: Is this really a continuous flow along the coast? Isn't this the track also of the NBC rings? There might appear a continuous flow in the average, but in practice the transport is accomplished by these large vortices, which are not wind-driven, but forced as you state by barotropic and baroclinic instabilities. Hence it is clear, that this does not follow the wind?!

Answer: Thank you for your comments. Indeed, you are right about the possible impact of the large vortices. However, to minimize their impact we have 4-month lowpass filtered the time series. Our results show a continuous flow, even though it is very weak except during boreal winter-spring period.

Concerning, the wind, we think that the wind has an effect. However, there also other parameters to be consider. We have then reformulated as follows: "Here, we also see that the seasonal cycles of the NBC branches north of the equator (except the NBC continuity along the Guyana coast) seems to follow the remote wind stress curl strength with a delay of one to four months (Fig. 5a-f). Because of the instabilities of the region, this delay should be impacted by the mesoscale activities and/or wave propagations in the region (Fonseca et al., 2004)."

Line 481: Here you mention the rings. I think you have to discuss this more clearly.

In general, all these acronyms make you go crazy, but I do not have a good suggestion about how to change this.

Answer: Thank you for your suggestion. We have considered it and have add to the paragraph: "The root mean square (rms) of the monthly mean values of its location (Fig. 5c) is nearly

constant (~2.3°), and is consistent with the evidence that there may not be a preferred season for NBC ring formation (Garzoli et al., 2003; Goni and Johns, 2003). Indeed, Garzoli et al. (2003) have shown using inverted echo sounder observations that there was a link between the rapid northward/southward extension of the NBC retroflection and the shedding of the rings in this region."

Line 504: What do you mean here? Eastward surface flow corresponds to the mixing of subsurface flow? I do not really understand what you want to say here.

Answer: Thank you for your question. We did not express well ourselves. The sentence has been reformulated as follows: "Urbano et al. (2008) showed with ADCP data at 38° W that the eastward NECC cycle may start in this region when the EUC is shallower and further north, and seems to be connected also to a shallower NEUC during this period of the year."

Line 519-520 nSEC mixes with NEUC? For my understanding the NEUC is a subsurface flow. Does it have a surface extension then at this time? Otherwise, I do not quite understand how nSEC and NEUC are supposed to mix. Please clarify.

Answer: Thank you for your question. To better understanding, we have reformulated our sentence as follows: "The presence of the sNECC flow in April-May may suggest that the NECC flow might be initiated by an eastward recirculation of the nSEC which flows on top of the NEUC during this period."

Line 529-530: Isnt that obvious? For my understanding you only have the nSEC because the Trades traverse the equator as the ITCZ is positioned always north of the equator. The seasonal migration of the ITCZ therefore mainly affects the seasonal cycles for the nSEC and the NECC and not as much for the other parts of the SEC or not? I think you should always discuss the seasonal cycles you obtain in conjunction with the changes in the large scale circulation.

Answer: Thank you for your comment. Yes, it obvious. We wanted to highlight the aspect of the circulation that you described, and also show that along the nSEC pathway that, there are some differences, especially west of 32°W where its seasonal cycle seems to be influenced by the cyclonic circulation during boreal spring (Which is discussed later in the discussion section). We have reformulated as follows: "Obviously, the cSEC and nSEC which are two branches of

the westward SEC do not have the same seasonal cycle (Fig. 4S4, S5, S6). This is due to the fact that, in the northern hemisphere, the nSEC is affected by the southerlies which cross the equator, inducing a migration ITCZ location. However, the cycles of the nSEC4, nSEC5 and nSEC6 have maxima at different periods of time. At 32° W, the nSEC (nSEC6) increases from April to reach a maximum of ~ 0.3 m s$^{-1}$ in August, following the migration of the ITCZ (Fig. 5b, f). During this time, at 42° W, the nSEC (nSEC5) migrates northward, and its intensity decreases until July, when it almost disappears. The eastward flow X then appears (Fig. 4S5). The nSEC5 is observed again after July, increases and reach a maximum of ~ 0.2 m s$^{-1}$ in March (Fig. 4S5, S6 and Fig. 5b)."

Line 530: the nSEC does not cross the S4 section at a right angle. Couldn't that also have imprints on not being able to correctly determine the seasonal cycle of the intensity? Okay this is mentioned in line 542. I would maybe point this out beforehand and in any way, I think for all this description maybe think about what is important to say and what is maybe not; in this case maybe do not consider the SEC part from S4?

Answer: You are right. This is why we did not mention the nSEC4 until the end of the section. However, knowing that it could draw the attention of any serious reader, we chose to notify why it shows this variability at the end.

Section 4.4: See my general remark. You have to clarify how trustworthy your geostrophic surface velocities are in close proximity to the equator and whether then this current "X" is real or also could be an artefact.

Answer: Yes, you are right. A response has been given above, after the general remark and been considered in the paper. We think that this geostrophic current product is trustworthy in close proximity to the equator. We also think that this current is real and may be even underestimated (See Fig. 2 above).

Figure 6 is more what I would envision for Figurer 5 on seasonal scales concerning a subplot for each current vein.
Maybe better to remove the seasonal cycle in Figure 6 though to better see interannual variations? Its rather hard to see anything beyond the seasonal signal in these time series; this

seems to have been done in Fig. 7 for some of the currents. Maybe somehow combine Figure 6 and 7?

Answer: Thank you for your suggestions. Because of the complex circulation of the tropical western boundary and its multiple currents, it was very difficult to make the figures simple and give all the information necessary as you have mentioned it before. We have tried to combine Figures 6 and 7 as suggested, but the resulting figure appears rather harder to understand. So, we chose to not combine them.

Line 592: What do the deviations stand for? Standard deviation? Standard error? Why do you state all these numbers in the text? What do we learn from that?

Answer: These deviations stand for the standard deviation. We found important to mention them because, it shows how the intensity/location of the currents varies over the time period of 25 years. And, this supports the importance of the interannual variability study.

Line 599: as mentioned above this should be clearly seen when the seasonal cycle is removed.

Answer: Yes, we can see it in Figure 7. Both Figures 6 and 7 could have been put together, but to avoid having much information in one figure, we have separated them.

Line 600: Isnt it obvious that they should be correlated, if the NBCR feeds the NECC? In any way which time series did you correlate. I thought you want to look at the relation on interannual time scales, but I think that a 3 month running mean does not remove the seasonal component. Along that lines before you say that interannual variability is only important in small region which should maybe imply that the correlations to the other current bands on interannual time scales must be very small? I think this needs clarification.

Answer: Thank you for your comment. As mentioned it in one of the responses above, we didn't want to say that there no interannual variability at all in the other regions. As we can see in Fig. 3 of the paper, there is clearly year-to-year variations. Here, the sentence also was not well formulated. But to not create confusion in the reader head, we have removed this sentence. Concerning the NBCR and the NECC, we know that the NECC partially fed by the NBCR, but does not totally depend on him (Bourlès et al., 1999a; Goes et al., 2005; Verdy and Jochum,

2005). So, we found it important to be mentioned. This information might mean that, in the near-surface layer, the NECC depend more on the NBCR. At this stage, we were only trying to highlight in the time series the variability from year-to-year to justify the importance of the interannual study and introduce the subject.

Line 617-619: Didn't you just say that the NECC and the NBCR were correlated and now no obvious relation can be seen. Or maybe you mean no relation between core position and strength is obvious? I think you have to rephrase this to make clear what is meant.

Answer: Thank you for your comment. We wanted to talk about the relationship between the position and the strength. It has been reformulated as follows: "No relationship was found between the intensity of the NECC branches or of the NBCR flows and their location."

Line 620: Would you expect such a relation necessarily? What would be the mechanism pushing the core to a certain position and in the same time influencing the intensity?

Answer: We were expecting a relationship between the NECC intensity and its core location. This would have confirmed that both the intensity and the core location are influenced by the wind stress curl strength. But, the fact that it is not the case might mean that the NECC intensity and its northward/southward location are not driven by the same mechanism at the interannual timescale.

Line 624: Here I find more methods/analysis are necessary, see my general comment above. You cite Hormann et al. 2012, where an extensive analysis of the interannual NECC variability and its relation to the Atlantic climate modes was made. Along that line: What are your new results about this? In Hormann et al. 2012 they use CEOF analysis, regression and composite analysis to determine this relationship.

Here only correlations of different timeseries is used, which I think is not sufficient as this extensive study about interannual NECC variability and its relation to the climate modes (Hormann et al. 2012) already exists.

Answer: Thank you for your comment. We understood your concern. As mentioned in the response of the general comments, because we are dealing with many currents, the goal was to look for the possible relationships in a simple way possible. This work also aims to give some

subjects to explore further on a case-by-case basis. As we are dealing with individual sections in different locations, and are considering the current strength/core location times series for our approach, the application of CEOF and its related regression and composite analysis could not be applied.

So, the spectral analysis has been done in our work to extract the signals that was needed, and we tried to look for the relationship between the time series by doing correlations with significative tests. The method used here to look for the relationship between the current strength/core location is similar the composite method. Knowing the proper period of the AZM and AMM events (lasting about three months), it is possible to correlate only these events with the 3-month means of the current anomalies (centered each month of the year) to see how the year-to-year variations are linked with. But as you suggested, since some possible relationships have been found, in the future studies to come on a case-to-case basis, we can consider sophisticated method as the CEOF to deepen the knowledges.

We have reformulated better the whole paragraph as follows: "To investigate the relationship between the AMM, the AZM, and the year-to-year variability of the characteristics of the different currents at different locations over 1993-2007 period, we computed the three-month average anomalies centered each month of both the time series of the anomalies of the currents and of the climate mode indexes (so-called 3-month anomaly time series). So, the AMM and AZM peak events (respectively, March-April-May and June-July-August) have been correlated with the 3-month anomaly time series of the currents in order to learn more about their possible relationship at the interannual timescale in the study area (Figures not shown). Only the correlations greater than $\pm 0.5$ and which have been found significant with 95 % of confidence level, performing the Student's test are discussed below (listed in Table 2)."

Line 625: I guess you mean 1993-2017?

Answer: Yes, indeed. Thank you for notification. It has been considered.

Line 628: What to you learn about the spatial characteristics of the interannual variability in the study area when you only correlate time series? See my comment above I think you need some sort of regression and composite analysis for that.

Answer: Thank you for your comment. The sentence was not well formulated. We have reformulated as follows: "So, the AMM and AZM peak events (respectively, March-April-May and June-July-August) have been correlated with the 3-month anomaly time series of the currents in order to learn more about their possible relationship at the interannual timescale in the study area". The whole paragraph is in the previous response.

Table 2: In the beginning of your study you show that interannual variability is basically only elevated in the NECC region. Please clarify for what kind of correlation you investigate the time series, this somehow does not get clear for me.

Answer: Thank you for your remarks and comments. They have been taken into consideration and the sentence have been reformulated to clarify things.

Line 701: I am not sure I understood everything. Maybe instead of all this neg/pos decrease/increase describe the relation in one way and then say vice versa?
But do I understand correctly that all these relationships between the currents and climate modes were already found and explained? So, what is new? Please highlight this more clearly.

Answer: Thank you for your comment and suggestion. There is a misunderstanding. Indeed, the study of the Atlantic climate modes in relation with the wind anomalies have been already explained by Cabos et al. (2019). However, the relation between the current and the climate mode is a new result. So, this paragraph is discussing the meaning of the findings in order to understand the probable mechanisms behind the relationships. The whole paragraph has been reformulated as follows: "Referring to Cabos et al. (2019), the relationships found between the currents and the AMM show the influence of the strengthening of the southeast trade winds on the southward migration of the nNECC core at 32°W, whereas the NBC intensity between 3°-5°S and the equatorial eastward flow X intensity west of 42°W decrease, and vice versa. Conversely, the strengthening of the southeast trade winds may influence the northward migration of the sNECC core at 32°W, whereas the NBC2, sNECC6 and nSEC6 intensities increase, and vice versa. Referring to the same authors, the relationship with the AZM indicate the probable influence of the positive westerlies anomalies in the western part of the basin on the negative anomalies of the sNECC and nSEC intensities at 42°W, and vice versa. Concerning the eastward flow X at 32°W the relationship with both the AZM and the AMM modes indicates its strengthening during the concurrent

events of positive westerlies anomalies in the western part of the basin and the negative southerlies winds anomalies, and vice versa."

Figure 8: I do not understand what is shown here in color. Probably you mean it is speed = sqrt (u^2+v^2) multiplied with the sign of the **zonal** velocity or not? If there is also the sign of meridional component in it would be confusing, but the plot looks like zonal?

Answer: Thank you for the remark. Yes, what is shown in colour is the speed multiplied with the sign of zonal velocity to see either the current is westward or eastward. We have reformulated this part of the caption as follows: "The velocity vectors are superimposed on the speed multiplied by the sign of their zonal components (m s$^{-1}$)."

Line 744-751: I think I do not understand this? So absolute velocities is something you do not have right. They should be something like Ekman+geostrophy, but again keep in mind that both a problematic at and close to the equator. Is this part then something like a validation that you compare this with other studies and say that they are maybe somewhat weak here? This could then also mean that your current X is not really trustworthy then? See my general remark about that.

Answer: As said in the introduction, our investigation on this eastward surface current was inspired by what was mentioned in the literature. We know that in the equatorial region, the dynamics is more complex. But, as shown at the beginning of our response, especially in Fig. 2 to 11 above, the observations clearly show an eastward flow at the surface. It is clear that there are other parameters to be considered. However, if we assume that about 78% of the variance of the ocean currents can be explained by the geostrophic and Ekman currents (Ralph and Nüler, 1999), we can be more optimistic. Fig. 2 shown here, also show that this product can reproduce approximately well the current at 12-m depth.
We have reformulated the sentences as follows: "In this study, the seasonal maps of the near-surface Ekman currents (figure not shown) showed westward currents with higher amplitudes in this equatorial band (larger than 0.2 m s$^{-1}$) compared to X, mostly east of 32°W. Then, we conclude that the near-surface velocities found here in the equatorial region should be underestimated, particularly in the eastern basin. This is consistent with the strengths of the X current found in Fig. 5b."

Line 752-755: but only the geostrophic component of this flow and again its maybe not well defined in this product.

Answer: As said in the beginning of our responses, the technic used in the equatorial region is based on the β-plane approximation of the Lagerloef methodology (Lagerloef et al., 1999) and the results seems to quite fit with the observations (Fig. 2 here) even if we don't have a longer time series of currents data. So, considering the results of the Fig. 2 here, we can give credit to the data, even if there are things to further deepen.

Fig. 10 so in this case the schematic is based on Fig. 8 as hopefully Figure 1 is based on 2a, but you need to keep in mind that you only show the geostrophic component! Or did you combine fIgure 8 and 9 to come up with Fig. 10? You can not really add the NBUC und EUC in this case as you have not included them in Fig. 8; you have the surface geostrophic currents, so the NBUC und EUC are not part of it. Why are there 3 arrows now for the NECC. I thought you divide into nNECC and sNECC what is the middle one for then?

Answer: We based Fig. 10 on Figure 9. But before, we have evaluated the importance of the Ekman currents on the geostrophic currents, and the results were almost the same. We didn't add the EUC. However, we thought that it is important to show the NBUC since it influences also the NBC cycle south of the equator. We leaved the size of the corresponding arrow the same, to not argue about it.

Line 841-843: Then maybe some regression for the individual branches should be performed here to see whether the results of Hormann et al. 2012 hold for the individual parts or not?

Answer: Thank you for your suggestion. As mentioned in the beginning of our responses, the goal was to analyze the spatial and temporal variability of the surface currents in a simple way possible since we are dealing with many currents. Our approach was to compute the strength of the currents and try to understand the relationships between them, and also with the large-scale wind variability and the tropical Atlantic climate modes. So, we have performed spectral analyses, then considered the signals mentioned in the introduction, and analyzed the relationships based on statistical tests. It is easier to do the analyses performed by Hormann et al. (2012) when you are concerned by the variability of one current. So, the authors have used

the complex EOF and the corresponding regression analyses on maps in the NECC domain to investigate the variabilities of the current.

Line 852: interannual variations only important in the Northeast I thought?

Answer: We thank you for your comments. We haven't expressed well our idea. It has been reformulated to avoid the confusion as shown in one of the answers above.

---

## Referee Report (RR1)

Re-Review of "Revisiting the tropical Atlantic western boundary circulation from a 25-year time series of satellite altimetry data" by Djoirka M. Dimoune, Florence Birol, Fabrice Hernandez, Fabien Léger, Moacyr Araujo [Paper # egusphere-2022-402]

The authors have generally corresponded well to most of my comments and suggestions. However, I still have a few concerns, which I think need to be addressed before considering this manuscript for publication.

First of all, I agree with the other reviewer that the choice of naming a current "X" is rather bad. I agree that the manuscript is already full of acronyms but finding one more for the "new" current does not make the change in my opinion and it would be much better to refer to some sort of … eastward surface current later than to X.

Secondly, while I strongly appreciate that the authors looked for other data sets to validate this product in the equatorial region, I think a bit more work has to be done on the section plots for the ADCP sections as I think they are now shown as part of the Supplementary Material. To my knowledge there are no shipboard ADCPs (sADCPs) which have data at the surface. You write that the first depth varies between 0 and 17m and an sADCP can not have data at 0m. You definitely have some blank bin and then depending on the bin size you have the first bin roughly I would say between 10-30m. In the plots you show you can also see that when the data is "at 0" it is just vertically pulled to 0m from somewhat like 17m like the others (you have these vertical lines in those plots). So, if your data set has identical values at 0m and 17m I would remove the 0m values as they give a wrong impression. What I also do not understand for your plots is that for the zonal component there is something odd with the colormap I think. How can you still have positive zonal velocities beyond the 0-contour line? For the meridional component it seems to work, but for the zonal plots something is strange.
And then you also have to take care in your formulation about the comparison as you can not really "compare" your velocities. You could maybe plot the surface velocities of your product on top of those sections and then you can "see" how well they fit. Whether an interpolation by eye from the ADCP data to your data seems sensible. I hope it gets clear what I am trying to say. What I envision I found e.g. in this paper:
Kopte, R., Brandt, P., Claus, M., Greatbatch, R. J., & Dengler, M. (2018). Role of Equatorial Basin-Mode Resonance for the Seasonal Variability of the Angola Current at 11°S, Journal of Physical Oceanography, 48(2), 261-281.
In their Fig. 1 geostrophic surface velocities from altimeter seem to be included in the plot of an ADCP. I think this would be a nice way of showing that these data sets somehow fit together.

My third suggestion is to recheck the language again. I found quite some mistakes in the third person with an "s" too much or missing. I am not quite sure whether I found all of them, so please give it a thorough read-through again.

Last, but not least I would like to suggest that you work a bit more on Figure 9. I gave some suggestions below how I think it could be improved a bit, because I think it's still quite full, which makes it a bit messy and hard to grasp. As this figure basically summarizes the result I think it should be a bit more easy to access.

**In addition to these general remarks, I have some detailed remarks throughout the text:**

**Line 19:** I would suggest to write characteristics instead of singularities.

**Line 22-23:** I am not sure whether there is a mistake here or maybe the phrasing is misleading : You say the maxima/minima occur within boreal winter/fall, when the ITCZ is at its southernmost/northernmost location.
Instead of using the boreal winter/fall expression which sounds like the ITCZ is at its southernmost/northernmost position directly within the next season, I would use the "months": The maxima/minima occur within February/August, when the ITCZ is at its southernmost/northernmost location.
At least the February/August definition is correct now for the ITCZ migration.

**Line 80-86:** I am still not quite sure about this paragraph. You start the introduction now as we agreed on that the WTA surface boundary circulation is a superposition of AMOC, STC and Sverdrup dynamics. In line 84 you now say the WTA surface circulation is wind-driven and highlight the components which are most important from the vorticity equation meaning Sverdrup. How does this fit together and secondly is this important here?

**Line 90-92:** Maybe rephrase this sentence. At least I understand that you want to say: The wind influence, which is related to the seasonal migration of the ITCZ, is reflected in the latitudinal shifts of the currents.

**Line 96:** , and also **on ..**

**Line 112:** AZM instead of ACM?

**Line 144-145:** , which uses the β-plane approximation as the Coriolis parameter vanishes close/at the equator.

**Line 149-150:** I think I would be a bit more careful here how you phrase the results on the comparison. You say satisfactory … okay maybe, if Lagerloef found similar comparisons and also called them satisfactory. The correlation might be of 0.71 for the zonal component alright, but the amplitude is a factor of 2 off most of the time. The meridional component is completely off showing opposite signs most of the time. So, I think you somehow also have to admit these deficiencies and make that clear in the phrasing of this sentence and then say but its similar to Lagerloef and hence probably okay as he called it satisfactory. By the way why do you show also the current meter data of 2019, if you have nothing to compare it with? Maybe then instead focus the plots on the comparison you have (2017-2018). An additional remark you could also compare the current meter data outside of your study area at e.g. 23°W or 10°W of the PIRATA buoys to check for the "equatorial consistency". Maybe there is more data overlap there.

**Line 153:** What do these numbers stand for? You say mean bias and standard deviation difference. What does that mean that you calculated here?

**Line 159:** Okay I see the point from above that's why you show the 2018/2019 part of the time series.

**Line 160-161:** Maybe rephrase this a little: "legitimizes our desire to know more about this flow". Somehow this sounds a bit unscientific.

**Line 183:** You used tau_x and tau_y in the equations now, but not in the text.

**Line 256:** I would add an introductory phrase to the sentence, something like:" To further investigate the variability of the different current branches, we then … "

**Line 272:** This is the first time that I encounter this "X" current again and I agree with the other reviewer that calling a current "X" is a really bad choice. Please reconsider this. I do understand that this paper is already full of acronyms making it difficult to read, but one more or less does not make the situation worse.

**Fig 3:** I appreciate the improvement of the figure. However, I would still rephrase the second sentence in the caption to something like: "On the left side of each panel the time average over the section is shown (thick line) framed by the corresponding standard deviation (thin lines). The color shading shows …." I think what is written is too complicated.

**Line 305:** paths -> path. The sentence starting with "These maxima …" is strange as it is now, please rephrase and make sure the brackets are set well.
**Line 305 ff:** I hope I understand now correctly: V_max = core velocity and location of V_max is core location;

strength/intensity time series is the average over the complete area between Vmax/2 and Vmax for every time step, correct?

Width of the current: Take the mean of the current over the section (left part of Fig.3) and check where it drops to zero. Do you take the average for that or you do that for every time step. I am not quite sure I understand; you come up with one number, so probably you do that for the average curve, but on the other hand you write about the time series for that, so please clarify.

**Line 329:** On the right side of each subplot …

**Line 336:** I do not see a monthly climatology of the current width in Fig. 5. See my comment above I also would not really know how this is obtained. In the end of the line: Our analyses …

**Line 366:** the NBC2 *intensifies* one month …

**Line 367:** This shows … contribution to the NBC  north of the equator.

**Line 374:** tends to retroflect …This can justify the phasing of the annual cycle of the …

**Line 387:** .. seem

**Figure 5:** the tick for April misses the "r". It only says "Ap"

**Line 423:** .. and starts decreasing …

**Line 444:** In my understanding these winds are called the south east Trades and not the southerlies. In addition, I think this sentence is somehow irritating in the aspect to what is the cause of the changes. The ITCZ position is always north of the equator due to the asymmetric SST distribution. The migration of the ITCZ follows the zenith of the sun, which moves between the turning circles. As the ITCZ is the position, where the trades meet they also show a seasonal variability with latitude. Please somehow clarify these things.

**Section 4.4:** I very much appreciate that you looked at other data sets trying to verify the eastward surface velocities with other observations giving trust to the geostrophic product. However, I think for the PIRATA velocities you need to be careful how you formulate the "good" agreement. Here you should really strike that it is probably underestimated in your product, as the amplitude of the eastward velocities in the PIRATA record are much bigger. However, you can also have ageostrophic velocities included in the PIRATA velocities, which are not in your product. This has to be clearly stated.

**Line 479:** Which characteristics do you mean, I think only the intensity and core velocity are considered here?

**Line 480**: I think its not so great to make the reader go back to end of section 3 to look up how you defined the intensity/core velocity of the currents. If I recall correctly you calculated something like mean over area where ($V\_max/2 < V < V\_max$) for the intensity and $V\_max$ as the core velocity in the latitudinal part of the section, where the current is located on average. So I would try to simply add something like that here.

**Figure 6/7:** I was really hoping that some of this could be combined.

**Line 517:** relationships

**Section 5:** How did you decide which current branches to show in the plots belonging to this section? Probably the ones from the area in Fig. 2 which show most pronounced interannual variability? Although no then you would probably only show the currents from section 6 and maybe section 5. So what drives your choice?

**Line 525:** 1993-2017

**Line 532:** Isn't it called the Student's t-test?

**Line 539-540:** I am not quite sure I understand. What is the difference between "insignificant" and "none"? Please clarify.

**Line 593:** Please consider my comments about these plots.

**Line 609/610:** This is exactly what I am saying about the ADCP observations and why you can not have data at 0m.

**Line 612-616:** This paragraph is still not very conclusive to me how it is written. Okay, you say Ekman velocities are westward. So to observe somewhat strong absolute eastward velocities you need quite strong geostrophic eastward velocities as v_abs = v_ekman + v_geo. Maybe that is meant with we conclude that the velocities found here (which are geostrophic) should be underestimated. Okay, but maybe rephrase a little to make it clearer if this is meant. But how is that consistent with the X current found in Fig. 5b? I do not understand what is meant with this.

**Line 619:** maximum -> maxima

**Line 631:** currents

**Figure 9:** The width of the arrows is proportional to the current amplitude you say. What is meant here? Do you mean V_max or intensity or really their amplitude which then should be something like the difference between max and min? What determines the length of the arrows then as the length also differs?
Why are the acronyms explained here in the caption again? In all the other figures the reader is referred to the acronym table. Why are there sometimes 1 and sometimes 3 arrows for the NEC (blue arrows)?
I still do not understand why in the JAS figure there are three arrows in the NECC region. The nNECC, sNECC and ? In some places there are 2-3 arrows after each other probably all representing the NECC and in other cases like for the NBC there are some arrows which cover quite some length extent. How did you decide about these things? Is there a meaning behind that? I think it would definitely help to have less arrows as it makes the plots quite messy, so maybe consider to reduce them, if there is no specific meaning behind the amount of arrows per current branch.

**Line 662/663:** Please rephrase, somehow the defined in the end sounds odd.

**Line 667:** on the one hand

**Line 687:** But -> However,

**Line 690/691:** You probably mean the eastern part of the region you are investigating. Eastern part of the basin could also be in front of Africa.

**Line 693:** … opens the way **for** further investigations

---

## Author Response (AR2)

**Response to the editor**

**Comments to the author**:

Dear Dr Dimoune,

Reviewer#1 is satisfied with the revised version of your manuscript but reviewer#2 still has a number of remarks, that require your attention.

I agree with the reviewers that "current X" is not the right way to refer to a current in an oceanography paper. I would suggest that you consult with specialists of the region (Bernard Bourles and Peter Brandt are examples, but of course you may think of others) to find the best name to describe this current in your manuscript, a name that would seem acceptable to a wide community and be likely to be picked up in future publications.

I wish you a happy new year,

Anne Marie Treguier

Answer:

Happy new year 2023 to you too Dr. Anne Marie Treguier. Thank you for accepting to be the editor of our paper and for your evaluation of this work.

We agree with your suggestion regarding the name of current X. It is important for this current to be more easily discussed by the wide community of oceanographers. After consultations with colleagues, we have decided to use in the new revised manuscript the name Equatorial Surface Current (ESC as an acronym). This name has been adopted by considering the position of the current (the equatorial region), and the fact that it flows in the upper layer of the ocean.

Regarding the reviewer 2, we thank him for his comments and suggestions. We have tried again to respond and clarify some misunderstandings. We hope to have taken into account all the suggestions to improve the manuscript and make it publishable in Ocean Science.

Regards,

Dr. Minto Dimoune

**Response to Reviewer 2**

**Re-Review of "Revisiting the tropical Atlantic western boundary circulation from a 25-year time series of satellite altimetry data" by Djoirka M. Dimoune, Florence Birol, Fabrice Hernandez, Fabien Léger, Moacyr Araujo [Paper # egusphere-2022-402]**

Answer: First of all, we want to thank you again for your review and for your comments which were very useful to improve the manuscript. We wish you a happy new year.

**The authors have generally corresponded well to most of my comments and suggestions. However, I still have a few concerns, which I think need to be addressed before considering this manuscript for publication.**

Answer: We are glad that we were able to give you most of the appropriate responses to your first review. We hope to do the same with your remaining concerns and to provide now a revised version of the paper that is suitable for publication.

**First of all, I agree with the other reviewer that the choice of naming a current "X" is rather bad. I agree that the manuscript is already full of acronyms but finding one more for the "new" current does not make the change in my opinion and it would be much better to refer to some sort of ... eastward surface current later than to X.**

Answer: Thank you for the comment. It has been considered, and after discussions with colleagues we finally adopted the name Eastward Surface Current which acronym is ESC in the new revised manuscript. We have also removed X in the figures and replaced it by ESC (Fig. 3, 4, 5, 6, 7 and 9).

**Secondly, while I strongly appreciate that the authors looked for other data sets to validate this product in the equatorial region, I think a bit more work has to be done on the section plots for the ADCP sections as I think they are now shown as part of the Supplementary Material. To my knowledge there are no shipboard ADCPs (sADCPs) which have data at the surface. You write that the first depth varies between 0 and 17m and an sADCP can not have data at 0m. You definitely have some blank bin and then depending on the bin**

size you have the first bin roughly I would say between 10-30m. In the plots you show you can also see that when the data is "at 0" it is just vertically pulled to 0m from somewhat like 17m like the others (you have these vertical lines in those plots). So, if your data set has identical values at 0m and 17m I would remove the 0m values as they give a wrong impression. What I also do not understand for your plots is that for the zonal component there is something odd with the colormap I think. How can you still have positive zonal velocities beyond the 0-contour line? For the meridional component it seems to work, but for the zonal plots something is strange.

And then you also have to take care in your formulation about the comparison as you can not really "compare" your velocities. You could maybe plot the surface velocities of your product on top of those sections and then you can "see" how well they fit. Whether an interpolation by eye from the ADCP data to your data seems sensible. I hope it gets clear what I am trying to say. What I envision I found e.g. in this paper:

Kopte, R., Brandt, P., Claus, M., Greatbatch, R. J., & Dengler, M. (2018). Role of Equatorial Basin-Mode Resonance for the Seasonal Variability of the Angola Current at 11°S, Journal of Physical Oceanography, 48(2), 261-281.

In their Fig. 1 geostrophic surface velocities from altimeter seem to be included in the plot of an ADCP. I think this would be a nice way of showing that these data sets somehow fit together.

Answer: Thank you for appreciating our efforts. As you write, there are no ADCP data in the first few meters below the surface because of the removal of the first bins, usually affected by noise. As you say, depending on the SADCP, its configuration and the parameters considered during the data processing, there is no data usually between 10-30 m-depth. It is what is represented on Fig. 2, 4, 5, 8, 9 and 10 of the supplement materials. Only in Fig. 3, 6 and 7 the currents were interpolated up to 0 m-depth but it was done before publishing the data and not by us (as mentioned in the manuscript you can find information about the data on the data center PANGEA website https://doi.pangaea.de/10.1594/PANGAEA.937809 following Tuchen et al, 2022). However, to be consistent with the other figures, and based on the product description we found, we have now hidden the first meters. The new figures are:

[Figure]

**Figure 3**: Shipboard ADCP section at 40°W between 2°S-2°N during 03/05/2003 to 05/05/2003 period. The dashed/solid contours represent the westward (northward)/eastward (southward) zonal (meridional) component of the current U (V). The contour intervals are each 0.2 m s⁻¹ for both components.

[Figure]

**Figure 6**: Shipboard ADCP section at 35°W between 6°S-8°N during 09/05/2002 to 16/05/2002 period. The dashed/solid contours represent the westward (northward)/eastward (southward) zonal (meridional) component of the current U (V). The contour intervals are each 0.2 m s⁻¹ for both components.

[Figure]

**Figure 7**: Shipboard ADCP section at 35°W between 5°S-3°N during 26/05/2003 to 01/06/2003 period. The dashed/solid contours represent the westward (northward)/eastward (southward) zonal (meridional) component of the current U (V). The contour intervals are each 0.2 m s$^{-1}$ for both components.

Regarding the 0-17 m-depth mentioned about the values in the first meters, it was a mistake in our response. For all of these sections, the starting depth varies from 17-30 m-depth. This is what should have been written. Thank you for your remark.

Thank you also for the paper (Kopte et al., 2018) you recommended. Regarding Fig. 1 of the authors, it was made by using an ADCP moored at 10°50'S, 13°E. It produced continuous data from August 2013 to October 2016. So, it was possible for the authors to compare the corresponding data with geostrophic currents after specific filtering. In our case, we have individual Shipboard ADCP (SADCP), which measurements are instantaneous at a few dates (then not continuous) and follow given transects. So, the comparison with gridded geostrophic current (resulting from an OI analysis of satellite data which are non colocated in time and space) at these dates is not obvious. However, the SADCP data at different seasons can be used to highlight the presence of the ESC (Equatorial Surface Current). In the different sections, the structure of the currents in the first meters (10-30 m-depth) can give an idea of whether there is an eastward surface current or not. It is clear that a surface current can usually be differentiated from a thermocline layer currents or Undercurrent. So, this was our approach in the manuscript when we mentioned it in the discussion (section 6).

Concerning the zonal component of the SADCP sections, we agree that Fig. 2, 3, 4, 8 and 10 of the supplement materials could be improved and it is what we did. Here are the new figures:

[Figure]

**Figure 2**: Shipboard ADCP section at 40°W between 2°S-2°N during 07/03/1994 to 10/03/1994 period. The dashed/solid contours represent the westward (northward)/eastward (southward) zonal (meridional) component of the current U (V). The contour intervals are each 0.2 m s⁻¹ (0.1 m s⁻¹) for the zonal (meridional) components.

[Figure]

**Figure 3**: Shipboard ADCP section at 40°W between 2°S-2°N during 03/05/2003 to 05/05/2003 period. The dashed/solid contours represent the westward (northward)/eastward (southward) zonal (meridional) component of the current U (V). The contour intervals are each 0.2 m s⁻¹ for both components.

[Figure]

**Figure 4**: Shipboard ADCP section at 35°W between 6°S-3°N during 30/05/1991 to 05/06/1991 period. The dashed/solid contours represent the westward (northward)/eastward (southward) zonal (meridional) component of the current U (V). The contour intervals are each 0.2 m s$^{-1}$ for both components.

[Figure]

**Figure 8**: Shipboard ADCP section at 35°W between 6°S-0°N during 17/10/1990 to 22/10/1990 period. The dashed/solid contours represent the westward (northward)/eastward (southward) zonal (meridional) component of the current U (V). The contour intervals are each 0.2 m s$^{-1}$ (0.1 m s$^{-1}$) for the zonal (meridional) component.

[Figure]

**Figure 10**: Shipboard ADCP section at 32°W between 2°S-2°N during 12/06/2006 to 13/06/2006 period. The dashed/solid contours represent the westward (northward)/eastward (southward) zonal (meridional) component of the current U (V). The contour intervals are each 0.2 m s⁻¹ for both components.

**My third suggestion is to recheck the language again. I found quite some mistakes in the third person with an "s" too much or missing. I am not quite sure whether I found all of them, so please give it a thorough read-through again.**

Answer: Thank you for the comment. We have made the necessary changes in the revised manuscript.

**Last, but not least I would like to suggest that you work a bit more on Figure 9. I gave some suggestions below how I think it could be improved a bit, because I think it's still quite full, which makes it a bit messy and hard to grasp. As this figure basically summarizes the result I think it should be a bit more easy to access.**

Answer: Thank you for your comment and suggestion. We did our best to improve Fig.9 and gave additional information in the text to help understand it. We hope you will find it better.

**In addition to these general remarks, I have some detailed remarks throughout the text:**

**Line 19:** I would suggest to write characteristics instead of singularities.

Answer: Thank you for your suggestion. It was made. See line 19 of the revised manuscript.

**Line 22-23:** I am not sure whether there is a mistake here or maybe the phrasing is misleading : You say the maxima/minima occur within boreal winter/fall, when the ITCZ is at its southernmost/northernmost location.
Instead of using the boreal winter/fall expression which sounds like the ITCZ is at its southernmost/northernmost position directly within the next season, I would use the "months": The maxima/minima occur within February/August, when the ITCZ is at its southernmost/northernmost location.
At least the February/August definition is correct now for the ITCZ migration.

Answer: Thank you for your suggestion. The months have been used in the previous version and it has been suggested to change them into the seasons because all the currents we were referring to don't always have maxima during the same months. However, we agree that the sentence was not well phrased. What we wanted to say is that, the maxima occur during boreal winter when the ITCZ is located equatorward; and the minima occur during boreal fall when the ITCZ is located northward. We rephrased the sentence as follows:
"with maxima/minima during late boreal winter/boreal fall when the Intertropical Convergence Zone is at its southernmost/northern location." (Line 22-23)

**Line 80-86:** I am still not quite sure about this paragraph. You start the introduction now as we agreed on that the WTA surface boundary circulation is a superposition of AMOC, STC and Sverdrup dynamics. In line 84 you now say the WTA surface circulation is wind-driven and highlight the components which are most important from the vorticity equation meaning Sverdrup. How does this fit together and secondly is this important here?

Answer: Thank you for your comment. It was not clear. We wanted to explain a little bit the dynamics of the region from the intraseasonal to the interannual timescales. But, as our study is focused only on the seasonal and interannual variability, the second sentence is not important here. So, we have removed it. Thank you for your careful review and remark.

**Line 90-92:** Maybe rephrase this sentence. At least I understand that you want to say: The wind influence, which is related to the seasonal migration of the ITCZ, is reflected in the latitudinal shifts of the currents.

Answer: Thank you for your comment and suggestion. The sentence was not clear. As you understood, we wanted to explain how the wind variability is related to the ITCZ migration and can also been linked to the shift of the currents. We have adopted you sentence and wrote it as follows: "This wind influence, which is related to the seasonal migration of the ITCZ reflects in the latitudinal shifts of the currents." (Lines 88-89)

**Line 96:** , and also **on ..**

Answer: Thank you for your comment. It has been corrected and formulated as follows: "and also on the seasonal cycle of the local wind forcing (Hisard and Hénin, 1987; Provost et al., 2004; Brandt et al., 2006; Hormann and Brandt, 2007; Brandt et al., 2016)." (Lines 93-95).

**Line 112:** AZM instead of ACM?

Answer: Of course, it is the AZM. But as the AZM is an Atlantic Climate Mode, we had chosen to say it in general point of view by saying ACM. However, thank you for your comment. We have changed it as follows: "They showed that the EUC transport is affected by the cold and warm events of the AZM (the so called "Atlantic Niño / Niña") and confirmed the previous findings of Goes and Wainer (2003) concerning the link between the interannual variability of the wind and the AZM impacting the strength of the tropical Atlantic circulation." (Lines 107-110).

**Line 144-145:** , which uses the β-plane approximation as the Coriolis parameter vanishes close/at the equator.

Answer: Thank you for your reformulation. It has been considered and the new idea is as follows: "In the equatorial band, the currents have been calculated using the Lagerloef methodology (Lagerloef et al., 1999) which used the β-plane approximation as the Coriolis parameter vanishes close/at the equator." (Lines 142-144).

**Line 149-150:** I think I would be a bit more careful here how you phrase the results on the comparison. You say satisfactory … okay maybe, if Lagerloef found similar comparisons and also called them satisfactory. The correlation might be of 0.71 for the zonal component alright, but the amplitude is a factor of 2 off most of the time. The meridional component is completely off showing opposite signs most of the time. So, I think you somehow also have to admit these deficiencies and make that clear in the phrasing of this sentence and then say but its similar to Lagerloef and hence probably okay as he called it satisfactory. By the way why do you show also the current meter data of 2019, if you have nothing to compare it with? Maybe then instead focus the plots on the comparison you have (2017-2018). An additional remark you could also compare the current meter data outside of your study area at e.g. 23°W or 10°W of the PIRATA buoys to check for the "equatorial consistency". Maybe there is more data overlap there.

Answer: Thank you for your comment. The sentence was not well formulated. We wanted to say satisfactory compared to the results of the previous studies in the equatorial Pacific (Picaut et al., 1989; Lagerloef et al., 1999). To avoid any confusion, it has been reformulated as follows: "The results of the comparison with the geostrophic currents interpolated to the same location were in agreement with the results of the previous studies in the equatorial Pacific (Picaut et al., 1989; Lagerloef et al., 1999)." (Lines 165-168).

Concerning the difference of amplitudes, as suggested, we added a sentence to notice that before talking about the correlation. The sentence is as follows: "However, in all these studies the geostrophic currents are underestimated compared to the observations, and this is because the contribution of the ageostrophic velocities has not been considered (e.g. Fig. 1 in the supplement)." (Lines 168-170).

The opposite signs that is shown on the graph of the meridional component sometimes is due to the underestimation of the amplitude of the geostrophic current. This is also the case of the previous studies cited.

Concerning the plots of the currents meter data of 2019, as you noticed in one of your comment below, we have explained it in Lines 159-160 of the previous revised manuscript (now Lines 273-275). Since one of our major result was the presence of the ESC, it is provided to show the surface current at this period.

Concerning the possible comparison with PIRATA buoys located outside our study area, we didn't find it necessary because the data were already validated in the study area. It shows at least that the geostrophic currents can be used to investigate the current variability in this region.

However, the validation of altimetry currents at all PIRATA buoy positions, especially in the equatorial region can be a good subject for another paper.

**Line 153:** What do these numbers stand for? You say mean bias and standard deviation difference. What does that mean that you calculated here?

Answer: As in Lagerloef et al. (1999) cited in our manuscript, the numbers are the differences (currents component and standard deviation) between the two measurements. The mean bias is the mean of the difference between the values of the two measurements while the standard deviation difference is the difference of the standard deviations of the two measurements. The sentence has been reformulated as follows: "The mean biases/differences in standard deviation for both components are respectively, 0.04/0.11 m s$^{-1}$ and 0.14/0.03 m s$^{-1}$." (Line 172-173)

**Line 159:** Okay I see the point from above that's why you show the 2018/2019 part of the time series.

Answer: Thank you. We have also mentioned it above.

**Line 160-161:** Maybe rephrase this a little: "legitimizes our desire to know more about this flow". Somehow this sounds a bit unscientific.

Answer: Thank you for your comment. We have rephrased it as follows: "These findings motivate our investigation of the variability of the surface currents in the equatorial part of our study area. (Lines 275-276)

**Line 183:** You used tau_x and tau_y in the equations now, but not in the text.

Answer: Thank you for your remark. It has been corrected. See line 186.

**Line 256:** I would add an introductory phrase to the sentence, something like:" To further investigate the variability of the different current branches, we then … "

Answer:  Thank you for your suggestion. We have reformulated as follows: "To further investigate the variability of the different current branches, we then monthly averaged the daily

geostrophic currents and extracted the cross-section geostrophic velocities along the six sections defined above (Fig. 1).” (Lines 257-259)

**Line 272:** This is the first time that I encounter this “X” current again and I agree with the other reviewer that calling a current “X” is a really bad choice. Please reconsider this. I do understand that this paper is already full of acronyms making it difficult to read, but one more or less does not make the situation worse.

Answer: Thank you for your comment. As we mentioned it at the beginning of the responses, we gave a name to the current X which is now the ESC (Equatorial Surface Current). We have also changed it in all the new revised manuscript.

**Fig 3:** I appreciate the improvement of the figure. However, I would still rephrase the second sentence in the caption to something like: “On the left side of each panel the time average over the section is shown (thick line) framed by the corresponding standard deviation (thin lines). The color shading shows ….” I think what is written is too complicated.

Answer: Thank you for your comment and suggestion. We have rephrased caption as you suggested. The new caption is as follows: “**Figure 3.** Hovmöller diagrams (1993 to 2017) of the cross-section current components (m s$^{-1}$) for S1, S2, S3 S4 S5 and S6. On the left of each panel, the time average over the section is shown (thick lines), framed by the corresponding standard deviation (thin lines). The color shading shows the northward/eastward (red) or southward/westward (blue) direction of the cross-current. Acronyms are listed in Table 1. The numbers next to the acronyms represents the number of the section. The green line in S1 indicates the time series of the maximum velocity of the cross-section current (NBC), and is framed by the time series of the maximum velocities divided by 2 (blue lines) over 1993-2017 period.” (Lines 311-317)

**Line 305:** paths -> path. The sentence starting with “These maxima …” is strange as it is now, please rephrase and make sure the brackets are set well.

Answer: Thank you for your correction and comment. The corrections have been made and the sentences are reformulated as follows: “In order to further investigate the temporal variations

of the current amplitude or strength, the maximum speed values of each current path have been determined for each month. The maximum speed corresponds to the current core velocity and is called Vmax hereafter: (see Fig. 3, green line on S1). The location of the core velocity is also computed." (Lines 318-321)

**Line 305 ff:** I hope I understand now correctly: V_max = core velocity and location of V_max is core location;
strength/intensity time series is the average over the complete area between Vmax/2 and Vmax for every time step, correct?

Anwser: Yes, it is correct.

Width of the current: Take the mean of the current over the section (left part of Fig.3) and check where it drops to zero. Do you take the average for that or you do that for every time step. I am not quite sure I understand; you come up with one number, so probably you do that for the average curve, but on the other hand you write about the time series for that, so please clarify.

Answer: Yes, we wanted to say the average curve over the period of study. To make ourselves more understandable, we have reformulated as follows: "The width of the currents (Table 2) is determined by the average curve over the whole period of study (1993-2017) (see left boxes of the Hovmöller diagrams of Fig. 3)." (Lines 324-326)

**Line 329:** On the right side of each subplot …

Answer: Thank you, it has been changed.

**Line 336:** I do not see a monthly climatology of the current width in Fig. 5. See my comment above I also would not really know how this is obtained. In the end of the line: Our analyses …

Answer: Thank you for your comment. Yes, you are right, it was a mistake. "Current width" does not has its place in this sentence. The sentence is reformulated as follows: "For further analysis, the monthly climatology of Vmax, Vmax location and the relative current intensity has also been derived from the corresponding monthly time series for different current components (Fig. 5)." (Lines 349-351)

The width of the current has been also clarified (see the previous response).

To avoid repetitions, we have removed the last sentence, and the whole paragraph became: "Here we focus on the seasonal cycle of the different current branches observed in the study area. Therefore, a monthly climatology of the velocity estimates shown in Fig. 3 is calculated for each of the six sections (Fig. 4). For further analysis, the monthly climatology of Vmax, Vmax location and the relative current intensity has also been derived from the corresponding monthly time series for different current components (Fig. 5)." (Lines 347-351)

**Line 366:** the NBC2 *intensifies* one month …

Answer: Thank you, it has been corrected (Line 381).

**Line 367:** This shows … contribution to the NBC to the north of the equator.

Answer: Thank you, it has been corrected and the sentence has been reformulated as follows: "This shows the importance of the nSEC contribution to the NBC in the northern hemisphere." (Lines 381-382).

**Line 374:** tends to retroflects …This can justify the phasing of the annual cycle of the …

Answer: Thank you, it has been corrected and the sentences have been reformulated as follows: "The latter may generate barotropic and baroclinic instabilities of the NBC which then tends to retroflect eastward. This could explain the phasing of the annual cycle of the NBC2, NBC4 and rNBC2." (Lines 386-388).

**Line 387:** .. seems

Answer: Thank you, it has been corrected (Line 401).

**Figure 5:** the tick for April misses the "r". It only says "Ap"

Answer: Thank you for you remarks. It has been corrected in the new revised version. Please, see also the corrected figure below.

[Figure]

**Line 423:** .. and starts decreasing …

Answer: Thank you for your correction. It has been corrected (Line 437).

**Line 444:** In my understanding these winds are called the south east Trades and not the southerlies. In addition, I think this sentence is somehow irritating in the aspect to what is the cause of the changes. The ITCZ position is always north of the equator due to the asymmetric SST distribution. The migration of the ITCZ follows the zenith of the sun, which moves between the turning circles. As the ITCZ is the position, where the trades meet they also show a seasonal variability with latitude. Please somehow clarify these things.

Answer: Thank you for your comment and suggestions. The south east Trades are also called southerlies in some papers (See Okumura and Xie, 2004; Snowden et Molinari, 2003; Marin et al., 2009, Simpson et al., 2018; ….).
However, to avoid any confusion we have considered your suggestion and have reformulated as follows: "This is due to the fact that, in the northern hemisphere, the nSEC can be affected by the Southeast Trades which cross the equator." (Lines 458-459).

**Section 4.4:** I very much appreciate that you looked at other data sets trying to verify the eastward surface velocities with other observations giving trust to the geostrophic product.

However, I think for the PIRATA velocities you need to be careful how you formulate the "good" agreement. Here you should really strike that it is probably underestimated in your product, as the amplitude of the eastward velocities in the PIRATA record are much bigger. However, you can also have ageostrophic velocities included in the PIRATA velocities, which are not in your product. This has to be clearly stated.

Answer: Thank you for your comment and suggestion. As you also mentioned it earlier, we have mentioned it in section 2.1 as follows: "However, the geostrophic components are underestimated, and this is because the contribution of ageostrophic velocities has not been considered (e.g. Fig. 1 in the supplement)." (Lines 168-170)
We have added to this paragraph (at the end): "Note that as mentioned in Sect. 2.1, the ESC amplitude captured in the altimetry product here is probably underestimated compared to the observations." (Lines 490-491)

**Line 479:** Which characteristics do you mean, I think only the intensity and core velocity are considered here?

Answer: Of course, it is the intensity and core velocity/location. You are right. Thank you for your comment. We have rephrased as follows: "The latter is analyzed here, using the time series of the characteristics (intensity and core velocity/location) of the different current branches (see end of Sect. 3) captured along the 6 sections (Fig. 6)." (Lines 494-496).

**Line 480**: I think its not so great to make the reader go back to end of section 3 to look up how you defined the intensity/core velocity of the currents. If I recall correctly you calculated something like mean over area where ($V\_max/2 < V < V\_max$) for the intensity and $V\_max$ as the core velocity in the latitudinal part of the section, where the current is located on average. So I would try to simply add something like that here.

Answer: Yes, it is correct. It has been already taking into account in one of the responses above. The sentences have been reformulated as follows: "The maximum speed corresponds to the current core velocity and is called Vmax hereafter: (see Fig. 3, green line on S1). The location of the core velocity is also computed. Then, to estimate the current relative strength/intensity, we have considered the part of the sections of velocities larger than Vmax/2 (see Fig. 3, blue

lines on S1), and we finally computed the values by averaging the velocity values over the area where V_max/2 < V < V_max.". (Lines 319-324)

Thank you again for your suggestion.

**Figure 6/7:** I was really hoping that some of this could be combined.

Answer: Yes, you are right. As you can imagine, we tried our best but to not create confusion in the head of the reader, we chose to have two figures instead of having one figure with too much information.

**Line 517:** relationships

Answer: Thank you, we have corrected it.

**Section 5:** How did you decide which current branches to show in the plots belonging to this section? Probably the ones from the area in Fig. 2 which show most pronounced interannual variability? Although no then you would probably only show the currents from section 6 and maybe section 5. So what drives your choice?

Answer: Thank you for your comment. Of course, we plotted the current branches of the area in Fig. 2 which show most pronounced interannual variability, investigating also the year-to-year variations found in the intensity, the core velocity/location of the current throughout the time period.

**Line 525:** 1993-2017

Answer: Thank you for your remark. It has been corrected. (Line 539)

**Line 532:** Isn't it called the Student's t-test?

Answer: Yes, you are right. It has been corrected. (Line 545)

**Line 539-540:** I am not quite sure I understand. What is the difference between "insignificant" and "none"? Please clarify.

Answer: Thank you for your comment. "insignificant" is for the correlation, and by "none" we wanted to say that no month is related to this correlation. We have clarified it as follows:" Where the correlations are found lower than 0.5, we indicate "insignificant" over the whole time period ("none" for no month is related to this correlation)." (Lines 553-556)

**Line 593:** Please consider my comments about these plots.

Answer: Thank you. We have considered it and new figures have been made (see the first responses above)

**Line 609/610:** This is exactly what I am saying about the ADCP observations and why you can not have data at 0m.

Answer: Yes, we agree with you. The explanations have been given in the first responses above, and we have considered your suggestions.

**Line 612-616:** This paragraph is still not very conclusive to me how it is written. Okay, you say Ekman velocities are westward. So to observe somewhat strong absolute eastward velocities you need quite strong geostrophic eastward velocities as v_abs = v_ekman + v_geo. Maybe that is meant with we conclude that the velocities found here (which are geostrophic) should be underestimated. Okay, but maybe rephrase a little to make it clearer if this is meant. But how is that consistent with the X current found in Fig. 5b? I do not understand what is meant with this.

Answer: Thank you for your comment. You are perfectly right. It was not clear. What we wanted to say after plotting the Ekman current in the whole tropical Atlantic (include the area outside of our study area), is that, the Ekman current seems to influence more the ESC in the central basin. But, it can create more confusion in the reader head. We have rephrased all the paragraph, removing this aspect and explaining now as follows: "In the equatorial region (2° S-2° N), Fig. 8 also shows the ESC with lower intensity. It appears to be extended east of 32° W and is stronger during boreal spring and fall. This feature can be related to the near-surface eastward flow mentioned previously by Hisard and Hénin (1987) and Bourlès et al. (1999b) on top of the EUC in the WTA. Hisard and Hénin (1987) explained the poor description of this current in the literature by the difficulty of ADCP measurements to fully capture the upper layer

currents in this area. They showed that this near-surface current, independent from the EUC, can reach amplitudes larger than 0.5 m s$^{-1}$ between 23° W and 28° W. Comparing their currents values to the ESC intensity in our study, we conclude that the weaker values found here (mostly towards 32°W: Fig. 5b) might be explained by the importance of the ageostrophic components of the currents which are not considered here." (Lines 619-628)

**Line 619:** maximum -> maxima

Answer: Thank you , it has been corrected (Line 631)

**Line 631:** currents

Answer: Thank you for your remark. We have corrected it (Line 643)

**Figure 9:** The width of the arrows is proportional to the current amplitude you say. What is meant here? Do you mean V_max or intensity or really their amplitude which then should be something like the difference between max and min? What determines the length of the arrows then as the length also differs?

Why are the acronyms explained here in the caption again? In all the other figures the reader is referred to the acronym table. Why are there sometimes 1 and sometimes 3 arrows for the NEC (blue arrows)?

I still do not understand why in the JAS figure there are three arrows in the NECC region. The nNECC, sNECC and ? In some places there are 2-3 arrows after each other probably all representing the NECC and in other cases like for the NBC there are some arrows which cover quite some length extent. How did you decide about these things? Is there a meaning behind that? I think it would definitely help to have less arrows as it makes the plots quite messy, so maybe consider to reduce them, if there is no specific meaning behind the amount of arrows per current branch.

Answer: Thank you for your comments and suggestions. Indeed, the arrows are proportional to the current intensity. It was a mistake and has been corrected. The arrows are wider when the intensity of the current is maximum (normal arrows) and decrease in the season they reach a minimum (dotted thin arrows). So, we have reformulated the sentences corresponding to the introduction of the figure and in the figure caption (Lines 646-647)

The intensity of the current is only defined here by the width of the arrows (not on the length) and depends on the location.

Since Fig. 9 is a major figure which summarizes the study, we chose to keep the acronyms to allow everyone to have a direct idea about what are they mean.

Also, since we followed the currents from one position to another in our study, we have found it good to put many arrows in order to show the variability of the intensity of the currents along their path. We have now mention it in the introduction to the figure (Lines 647-648).

Concerning the three arrows in the JAS figure, you are right. It had to be only two arrows: one for the sNECC and the other for nNECC. We have made a new figure with a new caption (see below).

[Figure]

**Figure 9.** Schematic view of the seasonal maps of the tropical western boundary surface circulation together with the subsurface NBUC signature at the surface. C1 and C2 represent the cyclonic circulations highlighted in this study. The size of the arrows is wider when the intensity of the current is maximum (normal arrows), and decreases with season to reach its minimum (dotted thin line). The NBUC is represented because of its contribution to the NBC transport and is shown by dashed arrows. The current branches which are already known are in orange/blue

(orange/blue for the current fed by southern/northern hemisphere water). The green arrows characterize the new branches observed. NBC is the North Brazil Current; nNBCR and sNBCR are the northern and the southern flows of the North Brazil Current retroflection, respectively; rNBC is the retroflected branch of North Brazil current; nNECC and sNECC are the northern and the southern branches of the North Equatorial Countercurrent (NECC), respectively; cSEC and nSEC are the central and the northern branches of the South Equatorial Current (SEC), respectively; and ESC is the Equatorial Surface Current.

**Line 662/663:** Please rephrase, somehow the defined in the end sounds odd.

Answer: Thank you for your suggestion. We have rephrased as follows: "To do so, a new approach based on the calculation of the current intensity was adopted, using six defined horizontal sections." (Lines 679-680)

**Line 667:** on the one hand

Answer: Thank you for your remark. We have corrected it (Line 684)

**Line 687:** But -> However,

Answer: Thank you for your suggestion. It has been considered and the sentence reformulated as follows: "However, the ageostrophic velocities need also to be considered here to fully understand the surface circulation." (Lines 704-705)

**Line 690/691:** You probably mean the eastern part of the region you are investigating. Eastern part of the basin could also be in front of Africa.

Answer: Thank you for your remark. You are right. We were meaning the eastern part of the region investigated in our study. It has been reformulated as follows: "The interannual variability is much weaker than the seasonal one. It is more important in the eastern part of our study area and there is no obvious regional pattern of low frequency variations." (Lines 706-708)

**Line 693:** … opens the way **for** further investigations

Answer: Thank you, it has been corrected. (Line 710)

---

## Author Response (AR3)

**Response to the Editor**

Dear Dr Dimoune,

Thanks for the careful revision of your manuscript. There are now only some technical details to correct, as indicated by the reviewer.

Your sincerely,

Anne Marie Treguier

Answer:

Thank you Dr. Anne Marie Treguier for your evaluation of this paper. Thank you also to the reviewer 2 for his review and comments. We have considered all the new comments and have made the corrections accordingly.

Regards,

Dr. Minto Dimoune

**Response to reviewer 2**

Dear all,

I only have a few minor/technical comments, then the paper is ready for publication in my opinion. I used the track changes version for the line numbers.

Answer: Thank you again for your review which has greatly helped to clear up misunderstandings and improve this paper. We are grateful.
We have considered all your minor/technical comments.

Details:
Line 42: Is the Atlantic Zonal Mode (AZM) meant here?

Answer: Thank you for your comment. We did not formulate well what we wanted to say. We wanted to say both the meridional and zonal modes. To make ourselves more understandable, we have reformulated the last two sentences as follows: "At 32°W, the interannual variability of the North Equatorial Countercurrent and of the northern branch of the South Equatorial Current (in terms of both strength and/or latitudinal shift) are associated to the Atlantic Meridional Mode whereas the Equatorial Surface Current intensity is associated to both the Atlantic Meridional and Zonal Mode phases.". (Lines 38-42)

Line 121: remove "us"

Answer: Thank you so much for the remark. We have removed it. (See line 117 in the revised manuscript)

Line 182: add "time period" before the numbers

Answer: Thank you for your comment. It has been considered. (See lines 164-165 in the revised manuscript)

Line 186:add "all" However, in all ....

Answer: Thank you for your comment. It has been considered. (See line 169 in the revised manuscript)

Line 187: compared to "direct velocity" observations

Answer: Thank you for your comment. It has been considered. (See line 170 in the revised manuscript)

Line 191: Substitute "The" with "Similar"

Answer: Thank you for your comment. It has been considered. (See line 174 in the revised manuscript)

Line 286: Substitute in with "to"

Answer: Thank you for your comment. It has been considered. (See line 265 in the revised manuscript)

Line 408: tends ->tend

Answer: Thank you for your correction. It has been considered. (See line 388 in the revised manuscript)

Line 457: add "the" before ESC

Answer: Thank you for your comment. It has been considered. (See line 435 in the revised manuscript)

Line 523: remove "the" before light

Answer: Thank you for your correction. It has been removed. (See line 498 in the revised manuscript)

Line 547: remove closing brackets

Answer: Thank you for your comment. We have removed it. (See line 522 in the revised manuscript)

Line 553: varie --> vary

Answer: Thank you for your correction. It has been considered. (See line 527 in the revised manuscript)

Line 559: change value to velocity

Answer: Thank you for your correction. It has been considered. (See line 533 in the revised manuscript)

Line 583: Correlations --> Correlation

Answer: Thank you for your correction. It has been considered. (See line 555 in the revised manuscript)

Line 633: close --> closing

Answer: Thank you for your correction. It has been considered. (See line 603 in the revised manuscript)

Line 638: maybe better to say "Also suggest"; confirm would be if you really have surface observations, here it is suggested as it looks like the are probably extended towards the surface

Answer: Yes, you are right. We have replaced "confirm" by "also suggest" (See line 609 in the revised manuscript). Thank you for your suggestion.

Line 686: amplitudes --> intensities
More --> Several

Answer: Thank you for your corrections. They have been considered. (See line 650 in the revised manuscript)